# A secretome atlas of cardiac fibroblasts from healthy and infarcted mouse hearts
Jasmin Bahr[1], Gereon Poschmann [2], Andreas Jungmann[3], Martin Busch[3], Zhaoping Ding[1], Jens Vogt[4], Ria Zalfen[1], Julia Steinhausen[1], Arlen Aurora Euan Martínez[1], Thorsten Wachtmeister [5], Daniel Rickert[5], Tobias Lautwein[5], Christina Alter[1], Junedh M. Amrute [6], Kory J. Lavine [6], Karl Köhrer [5], Bodo Levkau [4,7], Patrick Most[3], Kai Stühler[2,8], Julia Hesse [1,7] ✉ & Jürgen Schrader [1,7] ✉

Cardiac fibroblasts (CF) are key players after myocardial infarction (MI), but their signaling is only incompletely understood. Here we report a first secretome atlas of CF in control (cCF) and post-MI mouse hearts (miCF), combining a rapid cell isolation technique with SILAC and click chemistry. In CF, numerous paracrine factors involved in immune homeostasis are identified. Comparing secretome, transcriptome (SLAMseq), and cellular proteome disclose protein turnover. In miCF at day 5 post-MI, significantly upregulated proteins include SLIT2, FN1, and CRLF1 in mouse and human samples. Comparing the miCF secretome at days 3 and 5 post-MI reveals the dynamic nature of protein secretion. Specific in-vivo labeling of miCF proteins via biotin ligase TurboID using the POSTN promotor mirrors the in-vitro data. In summary, we identify numerous paracrine factors specifically secreted from CF in mice and humans. This secretome atlas may lead to new biomarkers and/or therapeutic targets for the activated CF.

Cardiac fibrosis is a common feature of ischemic heart disease and several other disease states, such as diabetes mellitus, and aging[1,2]. Fibrosis increases myocardial stiffness, thereby impairing cardiac function, which ultimately progresses to end-stage heart failure. Numerous studies emphasized, that the severity of cardiac fibrosis in patients correlates with adverse cardiac events and mortality[3]. However, no treatment is presently available to prevent excessive fibrosis.

Cardiac fibrosis is defined as an increase in myocardial extracellular matrix (ECM) deposition by cardiac fibroblasts (CF)[4], in that the equilibrium between synthesis and degradation of the individual ECM components is disturbed. Proteomic analysis revealed that 90% of cardiac ECM is composed of ten different proteins, including collagens (collagens I, III, and IV), non-collagenous glycoproteins (fibronectin and laminins), proteoglycans, glycosaminoglycans, and elastins[5]. A fundamental feature of the ECM is that it is altered during inflammation, injury, or infection and with age[6]. It is now being appreciated, that CF not only secrete ECM components, that

provide dynamic tissue organization and integrity, but also signaling molecules participating in many biological functions[7]. The endogenous mechanisms that restrain pro-fibrotic signals to protect the myocardium from progressive fibrosis are presently unknown.

Recent single-cell transcriptomics data suggest that CF express various paracrine factors which may signal to surrounding cells[8,9]. To this end, multiple secretory pathways (classical exocytosis and unconventional protein secretion (UPS)) have evolved for optimal precision and timing of intercellular dialogs[10]. Despite its importance in cell-cell interaction, the individual proteins which are secreted from MI-activated CF so far have neither been fully defined nor has their communication with surrounding cells been entirely explored.

In previous studies, the release of bioactive factors from CF has been assessed mostly indirectly and documented the profound effects of CF-conditioned medium on cardiomyocytes[11,12]. Interestingly, mice ablated for PDGFRα⁺ fibroblasts better preserved cardiac function after MI as

[1]Department of Molecular Cardiology, Medical Faculty and University Hospital Düsseldorf, Heinrich Heine University Düsseldorf, Düsseldorf, Germany. [2]Institute for Molecular Medicine, Proteome Research, Medical Faculty and University Hospital Düsseldorf, Heinrich Heine University Düsseldorf, Düsseldorf, Germany. [3]Division of Molecular and Translational Cardiology, Department of Internal Medicine III, Heidelberg University Hospital, Heidelberg, Germany. [4]Institute of Molecular Medicine III, Medical Faculty and University Hospital Düsseldorf, Heinrich Heine University Düsseldorf, Düsseldorf, Germany. [5]Genomics & Transcriptomics Laboratory, Biological and Medical Research Centre (BMFZ), Heinrich Heine University Düsseldorf, Düsseldorf, Germany. [6]Center for Cardiovascular Research, Department of Medicine, Cardiovascular Division, Washington University School of Medicine, St. Louis, MO, USA. [7]CARID, Cardiovascular Research Institute Düsseldorf, Medical Faculty and University Hospital Düsseldorf, Heinrich Heine University Düsseldorf, Düsseldorf, Germany. [8]Molecular Proteomics Laboratory, Biological and Medical Research Centre (BMFZ), Heinrich Heine University Düsseldorf, Düsseldorf, Germany. ✉e-mail: julia.hesse@uni-duesseldorf.de; schrader@uni-duesseldorf.de

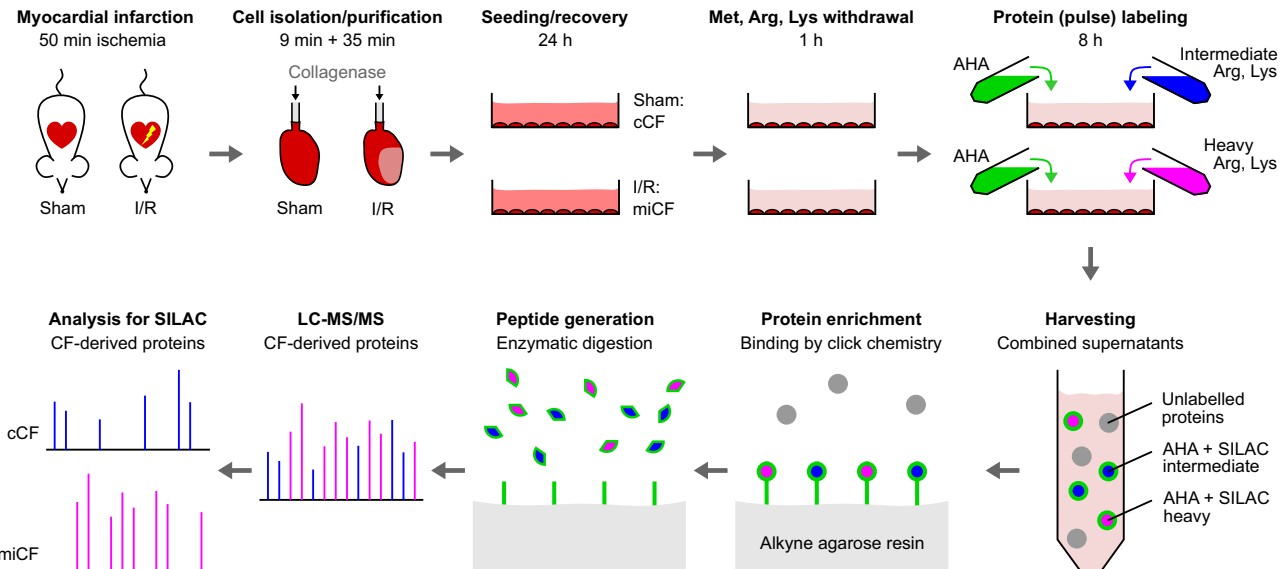

**Fig. 1 | Workflow of comparative secretome analysis of CF from sham control hearts (cCF) and infarcted hearts (miCF).** CF were isolated from mouse hearts 5 days after MI (50 min ischemia/reperfusion, I/R; miCF) or sham surgery (sham; cCF) and cultured in conventional cell culture medium containing 10% FBS for 24 h. Amino acids L-methionine (Met), L-arginine (Arg), and L-lysine (Lys) were withdrawn for 1 h and subsequently replaced by azidohomoalanine (AHA) together with either intermediate or heavy Arg and Lys isotypes ($[^{13}C_6]$ Arg, $[4,4,5,5-D_4]$ Lys or $[^{13}C_6, ^{15}N_4]$ Arg, $[^{13}C_6, ^{15}N_2]$ Lys, respectively). After incubation for 8 h, cCF and miCF supernatants were harvested and combined. Newly synthesized, AHA-containing proteins were bound to an agarose-resin by click chemistry. After stringent washing to remove non-bound proteins, remaining proteins were enzymatically digested and eluted peptides were applied to LC-MS/MS. In the following data analysis, intermediate and heavy isotype labels were used to assign peptides to cCF and miCF samples.

compared to wildtype controls[13], again suggesting that the secretome of CF is functionally important. However, up to now there is no study that has rigorously assessed the secretome of CF under in-vivo like conditions. All previous studies reported on the secretome of CF isolated from mice[14] and from human endomyocardial biopsies[15] were carried out under serum-free conditions, which very likely critically influence the cellular secretome[16].

The present study is the first to define the CF secretome in the non-ischemic and the ischemic heart using a combination of a rapid CF isolation technique together with stable isotope labeling using amino acids (SILAC) and click chemistry-based protein enrichment which permitted analysis under optimal, i.e., serum-containing, conditions. Aside of many proteins of the ECM, we identified numerous paracrine/autocrine factors that are specifically secreted from CF of healthy and infarcted hearts in mice and humans. The identified proteins in this secretome atlas may serve as a rich source to study in the future in more detail cell-cell interactions and/or to identify novel diagnostic markers for the infarct-activated fibroblast.

## Results

### Secretome analysis of cCF

For the comprehensive secretome analysis of CF isolated from the unstressed control heart (cCF) and the post-MI heart (miCF), we used a fast (9 min) cell isolation protocol[17] and short-term culture with 10% FBS, combined with click chemistry-based protein enrichment and SILAC labeling for robust protein quantification by LC-MS/MS[18] (Fig. 1).

In cCF isolated from sham control hearts we identified a total of 122 proteins, of which the majority of proteins (92%) were predicted to be secreted (Fig. 2A, Supplementary Data 1), either conventionally (Conv) or via UPS, indicating active release from cCF. In line with the key role of CF in maintaining structural tissue integrity, ECM-associated proteins accounted for about two thirds of the total protein amount (Fig. 2B). A major fraction of ECM proteins were collagen chains (Fig. 2C). We also found substantial secretion of proteins involved in collagen stabilization or turnover, such as lysyl oxidases (LOX, LOXL1-3), lysyl hydroxylases (PLOD1, 3), matrix metalloproteinases (MMP2, 3), as well as tissue

inhibitors of metalloproteinases (TIMP1-3), which together are likely to be involved in continuous matrix maintenance.

In addition to the ECM-associated proteins, the cCF secretome contained multiple proteins with potential paracrine and/or autocrine function (Fig. 2D), which amounted to 36% of the total measured protein intensity (Fig. 2B). The most abundant paracrine factor was plasminogen activator 1 (PAI-1, *Serpine1*), representing 68% of paracrine/autocrine protein intensity (Fig. 2D) and 25% of the total secretome (Fig. 2F). PAI-1 has been shown to control cardiac transforming growth factor-β (TGF-β) production and loss of PAI-1 results in cardiac fibrosis[19]. Supplementary Table 1 summarizes secreted proteins with reported biological function. In addition to PAI-1, this includes gelsolin (GSN), secreted frizzled-related protein-1 (SFRP1) and slit homolog 2 protein (SLIT2) as well as proteins influencing the complement system such as pentraxin-related protein 3 (PTX3) and complement C3 (C3). Immune homeostasis may be influenced by tyrosine-protein kinase receptor UFO (AXL), chemokine (C-C motif) ligand 2 (CCL2), and superoxide dismutase 3 (SOD3). Angiogenesis may be promoted by insulin-like growth factor binding protein 4 (IGFBP4), angiopoietin-like 4 (ANGPTL4), beta-nerve growth factor (NGF), and pigment epithelium-derived factor (PEDF, *Serpinf1*). Finally, proteins were secreted that are known to confer cardioprotection such as IGFBP4, NGF, and SFRP1, as well as proprotein convertase subtilis/kexin 6 (PCSK6). Note that several of the above proteins have previously not been assigned to CF as a primary site of production, such as AXL, IGFBP4, and PCSK6. Together these data suggest that cCF secrete numerous biologically active proteins that can signal to a variety of surrounding cardiac cells to maintain tissue integrity and homeostasis under unstressed conditions.

To gain further insight into the dynamics of protein secretion/turnover from cCF, we compared our secretome data with corresponding intracellular proteome data (Fig. 2E, Supplementary Data 2) and transcriptome data (Supplementary Data 3) that were generated by SLAMseq (see Methods). The graphical display of combined data in Fig. 3 enables a direct comparison of the secretome, proteome and transcriptome for all proteins secreted by cCF. It can be seen that PAI-1 (*Serpine1*), in contrast to its high secretion (Fig. 2F), amounted to only 0.4% of the intracellular intensity of secreted proteins (Fig. 2E). Together with substantial PAI-1 transcript levels and a

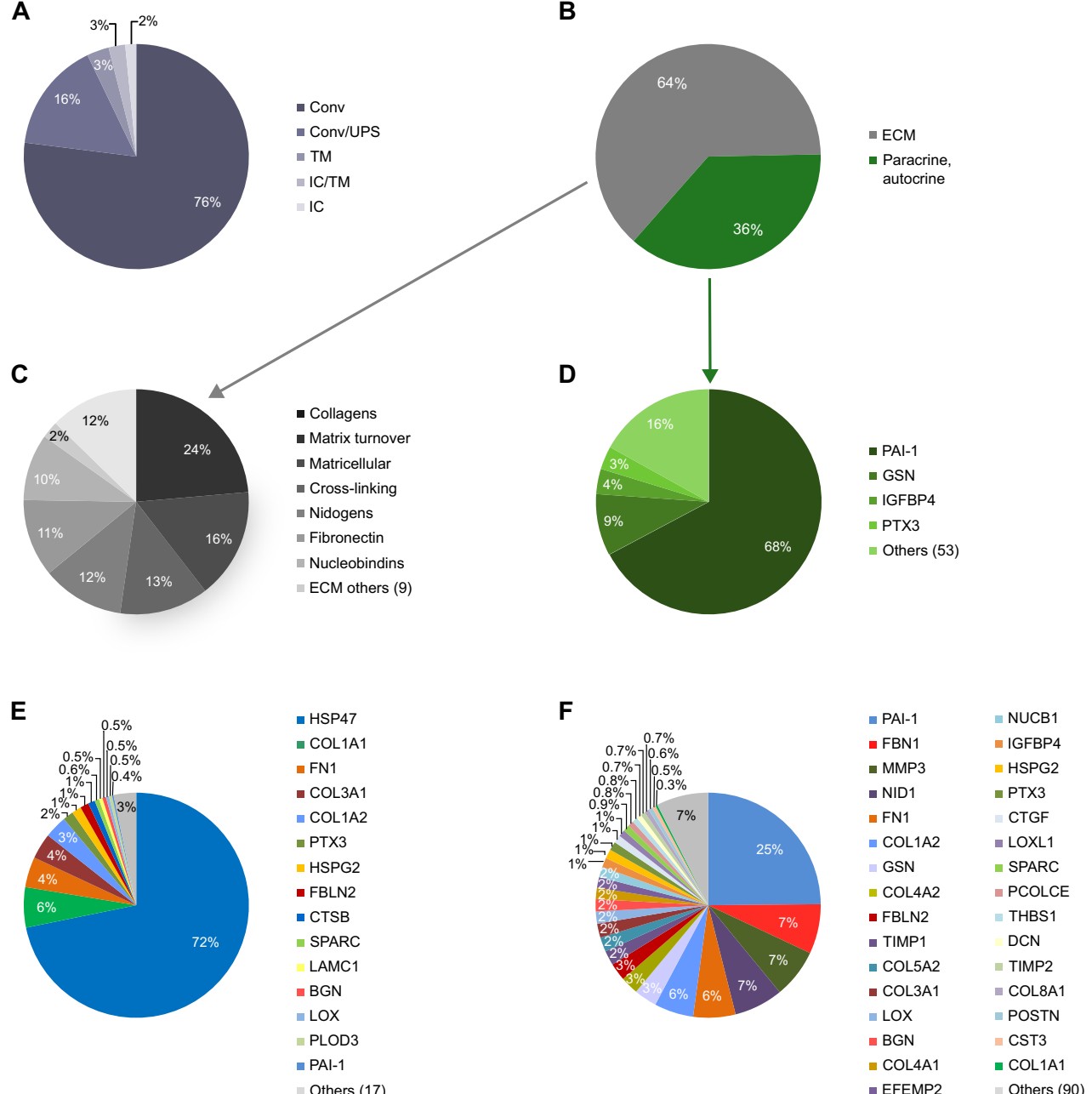

**Fig. 2 | Secretome analysis in cCF.** LC-MS/MS data identified 122 secreted proteins in cCF isolated from sham control mouse hearts 5 days after surgery (*n* = 4; source data in Supplementary Data 1). **A** Protein localization as predicted by OutCyte[62]. Shown is the percentage of released proteins classified as conventionally secreted (Conv), unconventionally secreted (UPS), intracellular (IC), and transmembrane (TM) proteins. **B** Protein intensity fractions of cCF secretome categorized as ECM-associated proteins and paracrine/autocrine factors according to KEGG annotation and PubMed search. **C** Subcategories of ECM-associated proteins. **D** Subcategories of proteins assigned as paracrine/autocrine factors. **E** Relative intracellular levels (intensities) measured in cell lysates of proteins that were identified in cCF secretome analysis (32 of the 122 secreted proteins). **F** Protein intensity representation of all proteins secreted from cCF.

high T>C conversion rate indicating active RNA synthesis (Supplementary Table 2), this suggests rapid intracellular turnover and immediate secretion of this central suppressor of cardiac fibrosis. In contrast, heat shock protein 47 (HSP47, *Serpinh1*), which is a member of the serpin protein family involved in assembly of triple-helical procollagen molecules in the ER[20], accounted for 72% of the intracellular intensity of secreted proteins (Fig. 2E) and showed substantial transcription, but only low secretion (Fig. 3, Supplementary Table 2). Similarly, collagen chains COL1A1, COL1A2, and COL3A1 exhibited high intracellular protein levels as well as elevated mRNA transcript abundances, but their secretion seemed to be relatively low (Fig. 3, Supplementary Table 2). It is important to note that the observed

secretome ECM protein levels might be underestimated, given the tendency of these proteins to remain primarily attached to the cell culture plates[21]. Also note, that many of the proteins secreted from cCF were below detection limit in the intracellular proteome analysis (90 of 122 proteins), indicating small intracellular protein pool sizes and therefore high cellular turnover. This includes paracrine factors, such as insulin-like growth factor (IGF1) and the chemokine CCL2, also known as monocyte chemoattractant protein 1 (MCP1) (Supplementary Table 2).

To address cell-cell communication between secreted paracrine factors and target receptors on individual cardiac cell types, receptors for paracrine factors were selected from the ligand-receptor database of CellTalkDB[22] and

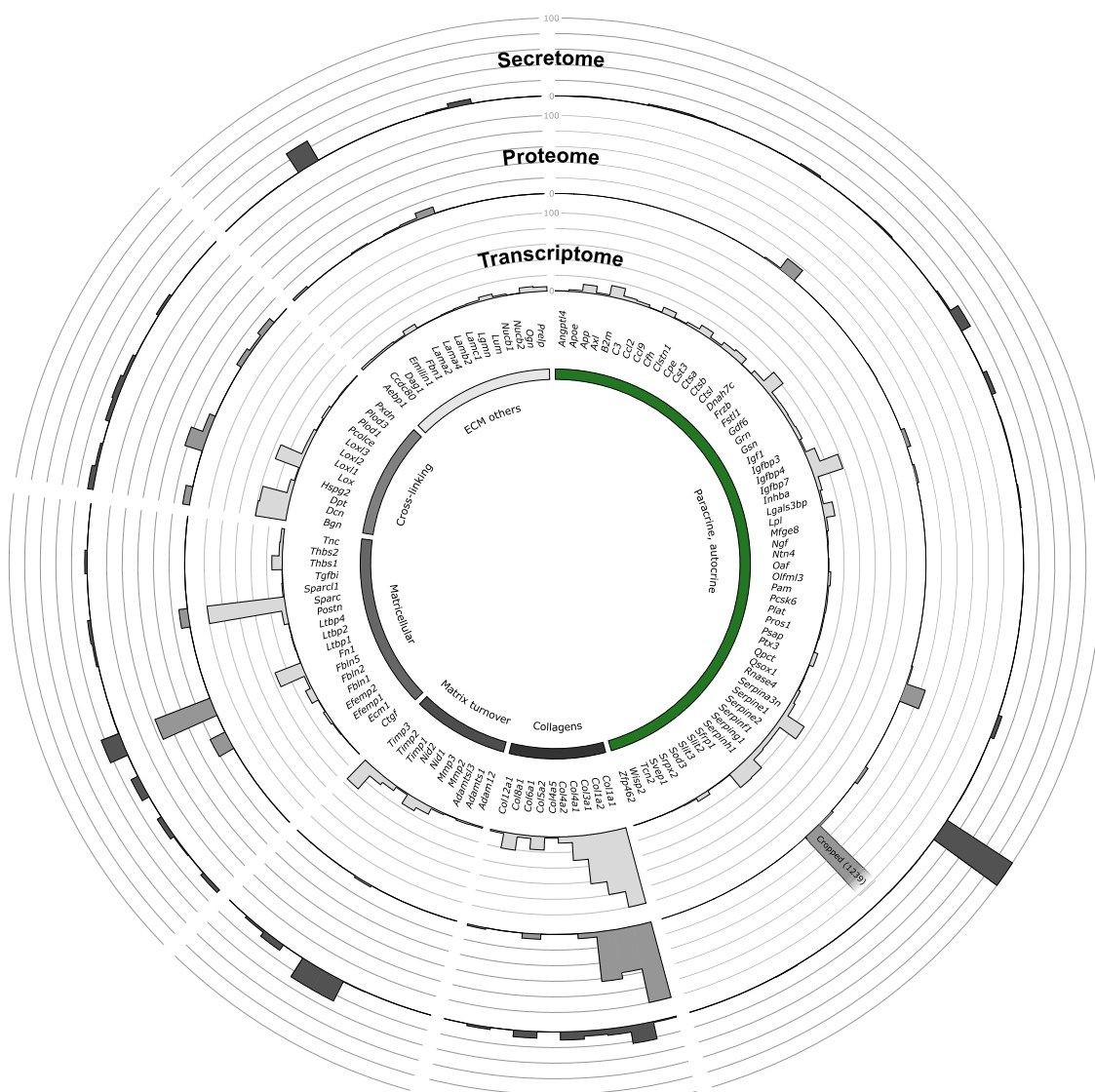

**Fig. 3 | Comparison of secretome, proteome, and transcriptome in cCF.** Gene expression (transcript levels, light gray bars; source data in Supplementary Data 3), cellular proteome (protein intensities, gray bars; source data in Supplementary Data 2), and secretome (protein intensities, dark gray bars; source data in Supplementary Data 1) of the 122 proteins identified in cCF secretome analysis. Data are shown in a Circos[64] plot as means of *n* = 3 (transcriptome) and *n* = 4 (secretome, proteome) of cCF preparations from unstressed hearts. To permit comparison, mean value of the gene/protein with the highest expression in the transcriptome (COL1A1, *Col1a1*) and secretome (PAI-1, *Serpine1*) was set to 100%. Due to the dominance of HSP47 (*Serpinh1*) in the proteome, the second highest value (COL1A1) was set to 100% and HSP47 (=1239%) was cropped to allow visualization of lower-abundant proteins.

their expression was analyzed in a publicly available snRNAseq data set of murine cardiac cells, including CM[23]. The interactome plot (Supplementary Fig. 2) revealed, that cCF-derived paracrine factors can signal to multiple cardiac cell types because of the broad expression of their respective target receptors.

To explore whether the secreted proteins are CF-specific, we searched a published murine cardiac scRNAseq data set[23] for the cellular distribution of the identified secretome proteins. As summarized in Supplementary Fig. 3 and 4, many of the proteins were found to be predominantly expressed by CF. Particularly GSN, highly secreted from cCF (Fig. 2D), showed high expression levels (Supplementary Fig. 4). On the other hand, C3 and SLIT3 were highly and preferentially expressed in epicardial cells. To further explore the translational potential of our findings in mice, we made use of recent snRNAseq and CITE-seq data sets generated by two of the coauthors (K.L. and A.J.) on human donor hearts, which were obtained from brain-dead individuals with no known cardiac disease and normal left ventricular function[24–26]. Note, that

samples for the human data sets were acquired from the left ventricular apex, while the whole heart was used for the mouse data set. Nevertheless, a direct comparison of the mouse with human data showed a remarkable similar distribution pattern at the gene expression level of the extracellular matrix proteins (Supplementary Fig. 3) and secreted paracrine/autocrine factors (Supplementary Fig. 4). Similar to mice, GSN was highly expressed in human CF and both C3 and SLIT3 transcripts were preferentially found in human epicardial cells (Supplementary Fig. 4).

### MI-induced changes in the CF secretome

To explore MI-induced changes in the CF secretome, we isolated activated fibroblasts (miCF) from infarcted mouse hearts (50 min I/R, 5 days after surgery). As shown in Fig. 4A, we identified 31 proteins which were only detected in the miCF secretome together with 122 proteins that were already present in the cCF secretome. Of all 153 identified proteins, 28 proteins were found to be significantly upregulated when comparing the miCF with the cCF secretome (Fig. 4B, Supplementary Data 1).

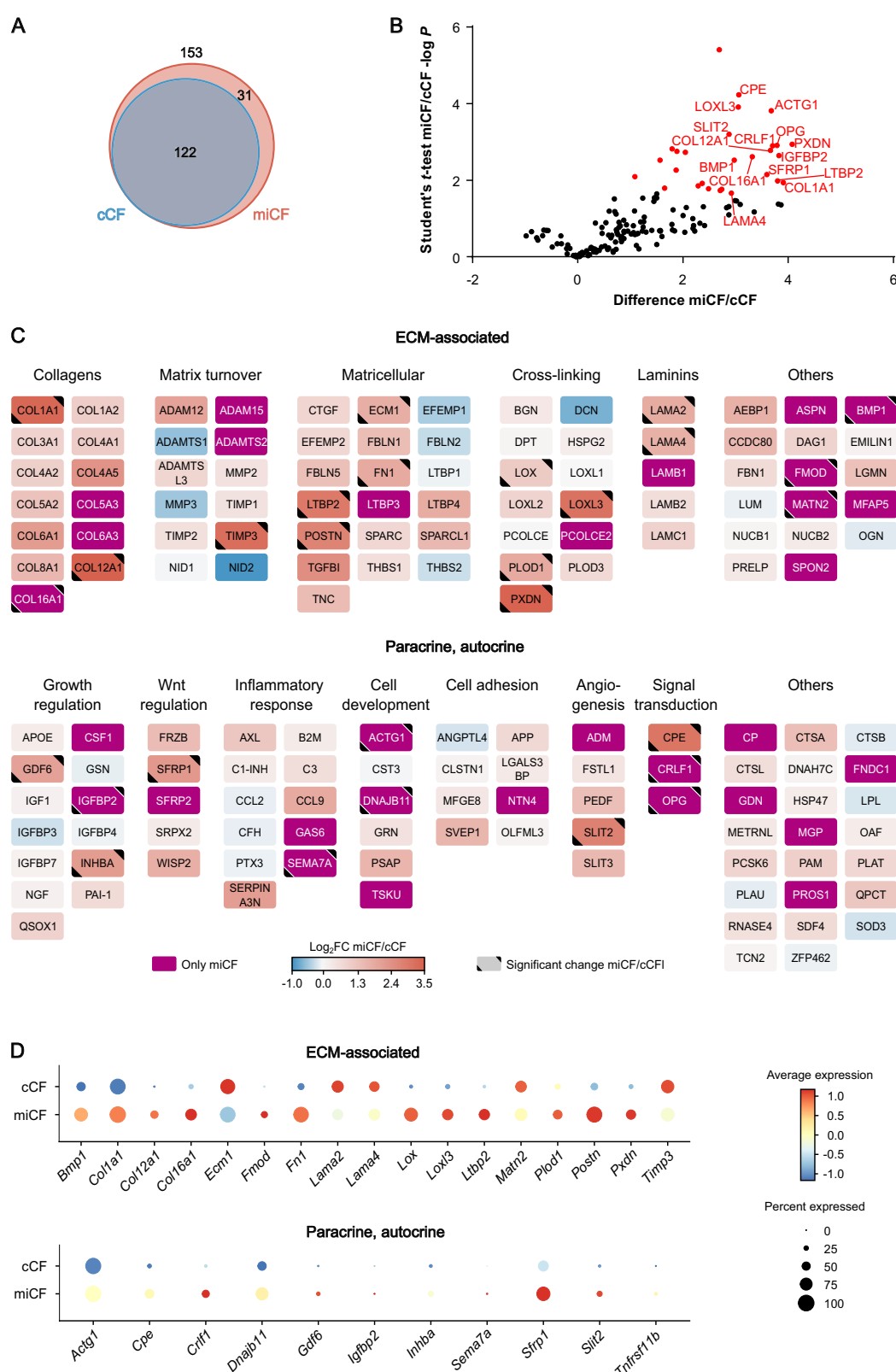

Figure 4C summarizes in a graphical manner the changes in the 153 proteins categorized into ECM-associated proteins and paracrine/autocrine factors. Some cardiac functions reported in the literature for the identified proteins are listed in Table 1. As can be seen, the ECM proteins that were found to be significantly enriched (Fig. 4C) included several proteins with known pro-fibrotic function, such as bone morphogenetic protein 1 (BMP1, only detected after MI), lysyl oxidase homolog 3 (LOXL3), periostin (POSTN), extracellular matrix protein 1 (ECM1), but also fibromodulin (FMOD, only detected after MI) with reported strong anti-fibrotic action. Levels of collagen chains COL1A1, COL12A1, and COL16A1 (only detected after MI) were also significantly enhanced (Fig. 4C). Increased proteins also included TIMP3, known to promote angiogenesis, as well as peroxidasin

**Fig. 4 | MI-induced changes of the CF secretome 5 days after infarction.** LC-MS/MS analysis identified 153 secreted proteins from post-MI CF (miCF) isolated from mouse hearts 5 days after I/R surgery ($n = 4$; source data in Supplementary Data 1). These data were compared to the basal secretome data (122 proteins) of cCF isolated from sham-operated hearts 5 days after surgery ($n = 4$; source data in Supplementary Data 1). **A** Venn diagram[65] of identified proteins. **B** Volcano plot of differentially secreted proteins. Proteins with significantly different intensities in cCF and miCF samples (Student's *t* test-based SAM analysis, 5% FDR, $S_0 = 0.1$) are highlighted in red (28 proteins). For statistical significance analysis of secreted proteins which were only detected in miCF, an imputation approach of missing base values was performed, using values taken from a downshifted normal distribution (details in Methods). Names of top 15 proteins with highest difference between cCF and miCF are annotated. The shown difference refers to the difference of group mean values of $\log_2$ transformed intensities. **C** Log2 fold changes (FC) of protein intensities between miCF and cCF samples visualized with Cytoscape[66]. Proteins were grouped in subcategories ECM-associated proteins and paracrine/autocrine factors. **D** Transcript levels of the 28 significantly changed proteins in the transcriptome of CF of mouse hearts 5 days after I/R (miCF) or sham surgery (cCF) ($n = 3$ each). Previously published scRNAseq data of cCF and miCF[8] were re-analyzed and expression levels in the total cCF/miCF population are visualized as dot plots.

**Table 1 | Reported functions of selected ECM-associated proteins and paracrine/autocrine factors significantly enriched in the miCF secretome as compared to the cCF secretome**

| | Protein; *gene* | FC | Function in cardiac ischemic stress response | Ref. |
|---|---|---|---|---|
| ECM-associated proteins | **Bone morphogenetic protein 1 (BMP1);** *Bmp1* | - | Promotes cardiac fibrosis | 77 |
| | **Fibromodulin (FMOD);** *Fmod* | - | Anti-fibrotic effects in cultured CF | 47 |
| | **Peroxidasin (PXDN);** *Pxdn* | 11 | Contributes to I/R-induced cardiac dysfunction | 78 |
| | **Tissue inhibitor of metalloproteinase 3 (TIMP3);** *Timp3* | 9 | Improves myocardial remodeling by promoting angiogenesis | 79 |
| | **Lysyl oxidase homolog 3 (LOXL3);** *Loxl3* | 8 | Promotes cardiac fibrosis by activating mTOR | 80 |
| | **Periostin (POSTN);** *Postn* | 6 | Participates in MI-induced fibrosis through CREB | 81 |
| | **Procollagen-lysine,2-oxoglutarate 5-dioxygenase 1 (PLOD1);** *Plod1* | 4 | Stabilizes collagen during fibrosis | 82 |
| | **Fibronectin (FN1);** *Fn1* | 4 | Promotes adverse cardiac remodeling, inflammation, and fibrosis | 28 |
| | **Extracellular matrix protein 1 (ECM1);** *Ecm1* | 3 | Leads to CF stimulation and fibrosis | 83 |
| | **Lysyl oxidase (LOX);** *Lox* | 2 | Contributes to adverse cardiac remodeling | 84 |
| Paracrine/autocrine factors | **Cytokine receptor like factor 1 (CRLF1);** *Crlf1* | - | Increases collagen production, CF proliferation, viability and myofibroblast transformation | 85 |
| | **Insulin-like growth factor binding protein-2 (IGFBP2);** *Igfbp2* | - | Prognostic biomarker for heart failure patients | 49 |
| | **Osteoprotegerin (OPG);** *Tnfrsf11b* | - | Elevated plasma levels in patients are associated with increase of cardiovascular events | 51 |
| | **Semaphorin 7A (SEMA7A);** *Sema7a* | - | Promotes thrombo-inflammation in MI | 86 |
| | **Carboxypeptidase E (CPE);** *Cpe* | 8 | Inhibits proliferation and apoptosis in primary cardiomyocytes | 87 |
| | **Slit homolog 2 protein (SLIT2);** *Slit2* | 7 | Inhibits inflammatory response and maintains myofilament contractile properties post-MI | 88 |
| | **Secreted frizzled-related protein 1 (SFRP1);** *Sfrp1* | 5 | Reduces fibrosis, inhibits CM apoptosis and improves cardiac function after MI | 48 |
| | **Activin A (Inhba dimer);** *Inhba* | 4 | Impairs human cardiomyocyte contractile function | 89 |
| | **Growth differentiation factor 6 (GDF6);** *Gdf6* | 4 | Enhances angiogenesis | 90 |

(PXDN) and fibronectin (FN1), which both have a reported pro-apoptotic activity in cardiomyocytes[27,28]. Note, that levels of several proteins like ADAM metallopeptidase with thrombospondin type 1 motif 1, MMP3, nidogen-2, EGF-containing fibulin-like extracellular matrix protein 1, thrombospondin 2, and decorin (DCN) were reduced in the secretome of miCF as compared to cCF (Fig. 4C), but differences did not reach the level of significance.

In addition to the 17 significantly enriched ECM-related proteins, we identified 11 significantly enhanced paracrine/autocrine factors in the miCF secretome (Fig. 4C). Among these were semaphorin 7A (SEMA7A; only detected after MI), which is known to mediate thrombo-inflammation, and growth differentiation factor 6 (GDF6), which can promote angiogenesis (Table 1). Notably, two proteins only detected in the miCF secretome, insulin-like growth factor binding protein 2 (IGFBP2) and osteoprotegerin (OPG), have already been reported as prognostic biomarker in human heart failure and additional cardiovascular events, respectively.

The list also includes several proteins with reported cardioprotective activity, such as carboxypeptidase E (CPE), Wnt/β-catenin pathway inhibitor SFRP1, and SLIT2. To explore whether these proteins might be of functional relevance, we assessed the anti-apoptotic activity of cell supernatants derived from cCF and miCF on isolated primary mouse cardiomyocytes, subjected to hypoxia to simulate ischemia/reperfusion (I/R) injury. While cCF-conditioned medium did not prevent I/R-induced cardiomyocyte cell death, cardiomyocyte cell viability was no longer significantly reduced in the presence of miCF-conditioned medium (Supplementary Fig. 5, Supplementary Data 6). This indicates that the secreted factors from miCF can effectively attenuate apoptosis/cell death in hypoxia-stressed cardiomyocytes.

To study whether the enhanced protein secretion by miCF is reflected by respective changes in gene expression, we analyzed scRNAseq data sets previously obtained by us under identical experimental conditions (CF isolated from sham control hearts and infarcted hearts 5 days post I/R)[8]. As displayed in Fig. 4D, 12 out of the 17 enhanced ECM-related secretome proteins and all of the enhanced paracrine factors also showed higher gene expression levels in miCF compared to cCF. This indicates that upregulation at the transcriptional level can explain the increased protein secretion.

To investigate, whether the identified 28 proteins that were significantly enriched in the miCF secretome are also predominantly expressed in CF in comparison to other cardiac cell types, we made use of scRNAseq data previously published by us[29] as well as data from acute MI patients (AMI, <3 months post-MI)[24–26]. As summarized in Fig. 5A, genes encoding

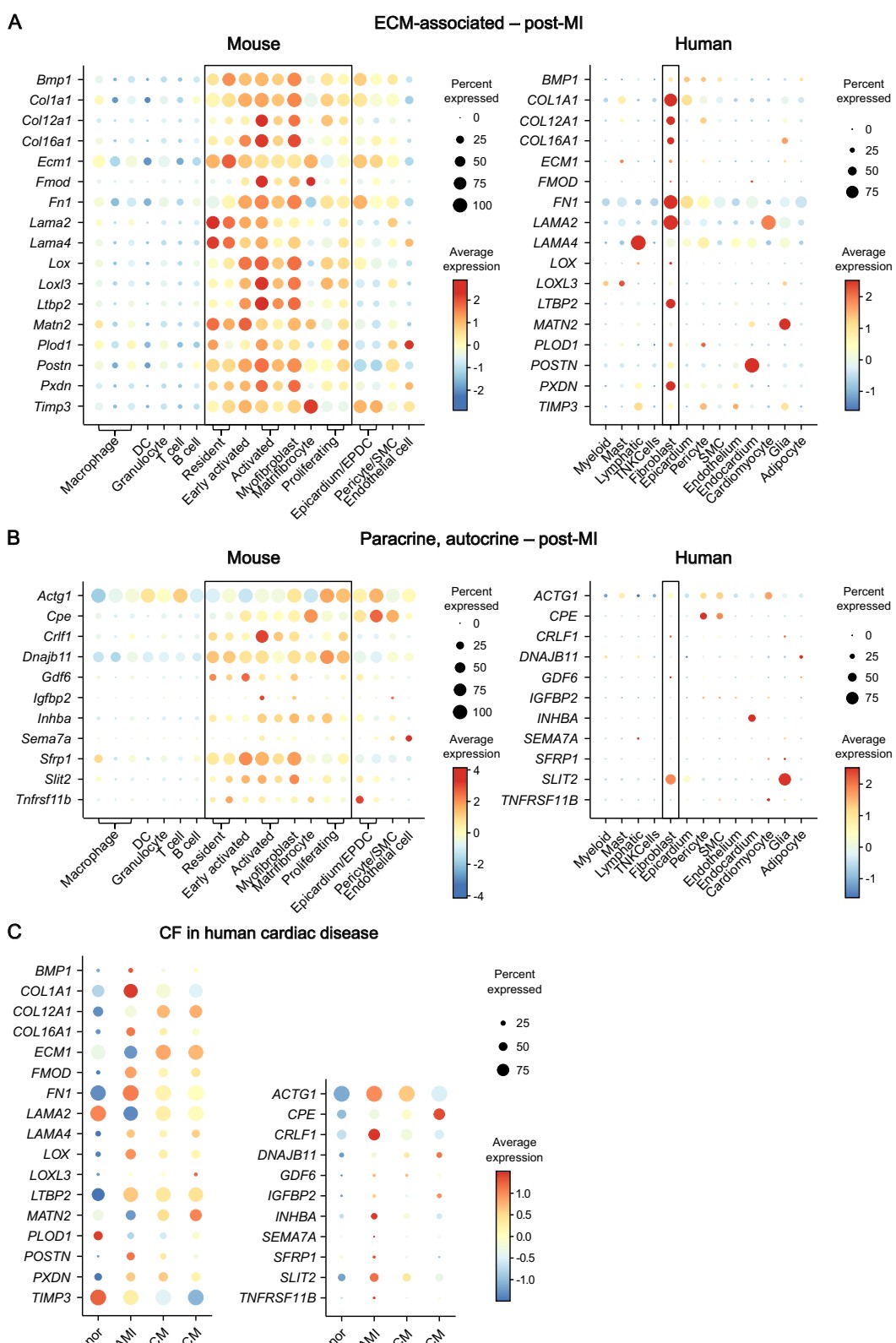

**Fig. 5 | CF specificity of significantly altered secretome proteins post-MI.**
**A**, **B** Transcript levels of the 28 significantly changed proteins between the miCF secretome and the cCF secretome in the transcriptome of cell populations of $n = 3$ infarcted mouse hearts 5 days after I/R (left panels) and of $n = 3$ human heart specimen of acute MI (AMI, <3 months post-MI) patients (right panels). Previously published scRNAseq data of murine stromal and immune cell populations[29] and snRNAseq data of human cardiac cells[24–26] were re-analyzed. Expression levels are visualized as dot plots, subdivided in (**A**) ECM-associated proteins and (**B**) paracrine and autocrine factors. Values of CF populations are enframed. DC, dendritic cell; EPDC, epicardium-derived cell; SMC, smooth muscle cell. **C** To compare the expression of the miCF secretome proteins in CF in human MI and heart failure, we used CITE-seq data[26] of human left ventricle (LV) CF obtained from $n = 6$ healthy donors, $n = 4$ acute MI (AMI) patients (<3 months post-MI), $n = 6$ ischemic cardiomyopathy (ICM, >3 months post-MI) patients, and $n = 6$ nonischemic cardiomyopathy (NICM, idiopathic dilated cardiomyopathy) patients. Details regarding the origin of tissue collected are provided in the method section.

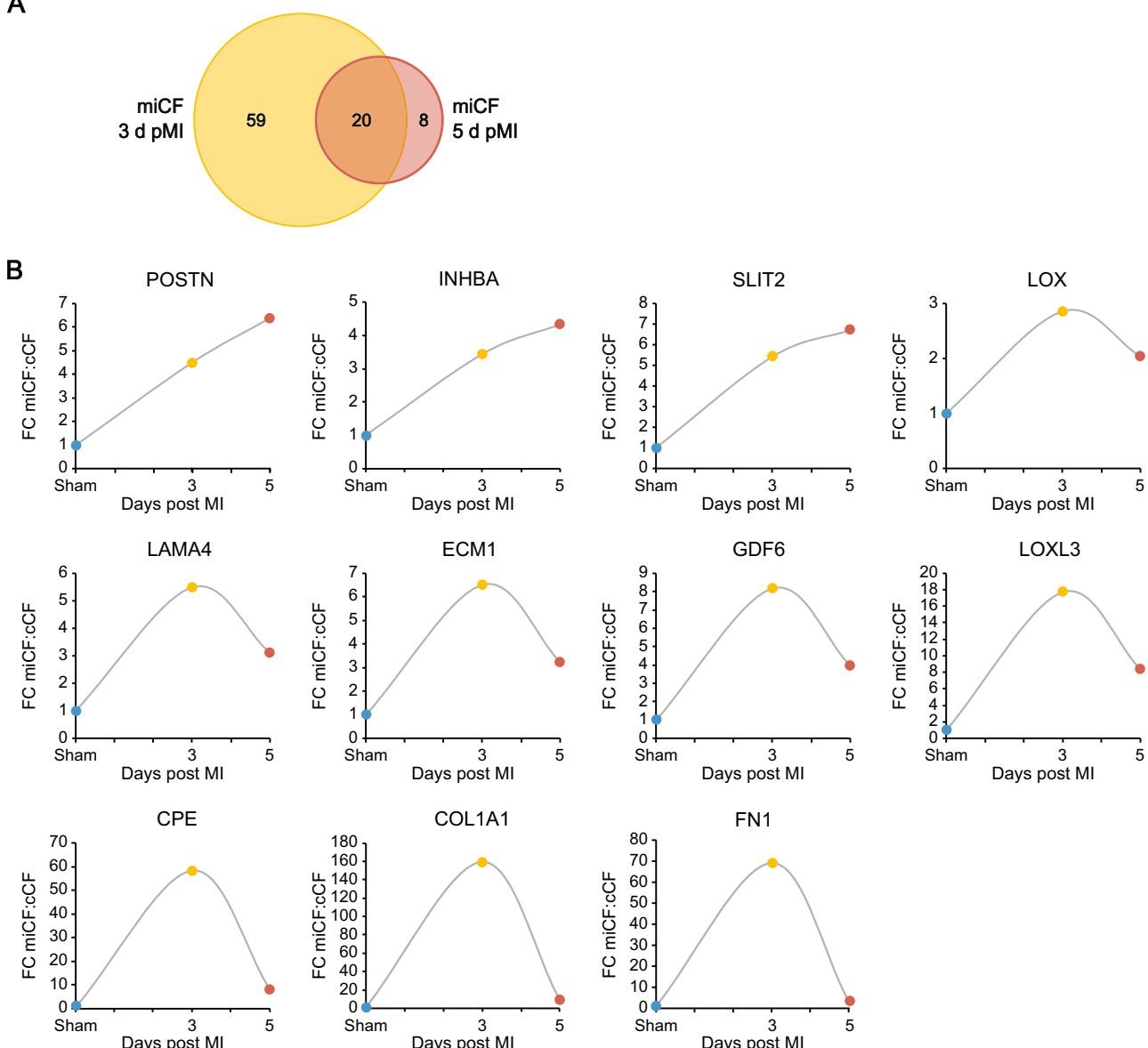

**Fig. 6 | MI-induced changes of the CF secretome 3 and 5 days after infarction.** LC-MS/MS analysis identified 129 secreted proteins from post-MI CF (miCF) isolated from mouse hearts 3 days after I/R surgery (n = 4; source data in Supplementary Data 1): These data were compared to the secretome data (153 proteins) of miCF from mouse hearts 5 days after I/R surgery (n = 4; source data in Supplementary Data 1). **A** Venn diagram[65] of significantly changed secretome proteins of miCF in comparison to the secretome proteins of cCF from sham-operated hearts 3 and 5 days after surgery (n = 4 each, source data in Supplementary Data 1), respectively. **B** Fold changes of selected secretome proteins between miCF and cCF at days 3 and 5 post I/R and sham surgery, respectively.

the enriched ECM-related secretome proteins and paracrine/autocrine factors were for the most part selectively expressed in miCF in mice and humans. Note, that the expression of the proteins was rather heterogeneously distributed within the different miCF populations in mice. The overlap between mice and humans was less pronounced for paracrine/autocrine factors (Fig. 5B).

The available human data set[24-26] contains besides CF samples from healthy individuals (donor) and AMI patients also samples from, ischemic cardiomyopathy (ICM, >3 months post-MI) patients and non-ischemic cardiomyopathy (NICM, idiopathic dilated cardiomyopathy) patients. This permitted us to explore whether the identified secreted proteins are infarct-specific or can also be observed in other heart pathologies such as ICM and NICM. As summarized in Fig. 5C, gene expression of most of the identified proteins was preferentially upregulated in AMI CF. ICM and NICM CF showed a different signature, with high expression levels of COL1A1, ECM1, matrilin-2, and CPE. Together these data indicate, that the

secretome of CF in the human heart is critically determined by the underlying cardiac disease.

To obtain insight into the temporal changes of protein secretion from MI-activated CF we carried out identical experiments as shown in Fig. 4 (day 5 post-MI), at day 3 post-MI (Supplementary Fig. 6, Supplementary Data 1). Comparing both time points, we found overall more proteins to be significantly changed in comparison to sham control on day 3 (79 proteins) than at day 5 post-MI (28 proteins), with an overlap of 20 proteins (Fig. 6A and Supplementary Table 3). Again, numerous ECM-associated proteins and paracrine/autocrine factors were significantly upregulated (Supplementary Fig. 6C) of which CCL9, granulin (GRN), LOXL2, osteoglycin (OGN), and SLIT2 were reported in the literature to be involved in fibrosis development. From the temporal changes (Fig. 6B) it can be seen that the secretion of some proteins increased over time (POSTN, INHBA, SLIT2) while the others peaked at day 3 and thereafter decreased to a different extent. Particularly, secretion of CPE, COL1A1, and FN1 dramatically

decreased after day 3. Together these findings are in line with the notion that CF are a very dynamic cell type that achieves selective differentiated states in the process from acute wound healing to long-term remodeling[30].

## TGF-β- and hypoxia-induced changes in the CF secretome and in-vivo-secretome analysis

To better understand the stimuli driving CF protein secretion, we investigated the impact of specific stressors associated with myocardial ischemic injury, TGF-β signaling and hypoxia, on the secretome of cultured CF isolated from unstressed hearts. TGF-β is a potent fibrogenic mediator, involved in repair, remodeling, and fibrosis of the injured heart[31]. As shown in Supplementary Fig. 7A, activation of TGF-β signaling (10 ng/ml recombinant mouse TGF-β1) in CF for 48 h induced significant changes in 22 secreted proteins in comparison to TGF-β signaling inhibition (10 μM TGF-β receptor inhibitor SB-431542) (Supplementary Data 1). Among the 14 proteins, that were significantly increased after TGF-β stimulation, 5 proteins have been identified to be significantly upregulated in the miCF secretome: LOX, FN1, INHBA, POSTN, and COL1A1 (Supplementary Fig. 7B). Some of the other significantly upregulated miCF secretome proteins were also detected in the TGF-β secretome analysis, but were not significantly altered between TGF-β stimulation and TGF-β inhibition samples (LTBP2, COL12A1, ACTG1) (Supplementary Fig. 7B), suggesting that additional stimuli are involved in the induction of miCF secretome proteins. Exposure of CF to hypoxia (1% O$_2$) for 8 h did not result in any significant changes in the secretome in comparison to CF incubated under standard cell culture conditions (20% O$_2$) (Supplementary Data 1). Also, proteins significantly enriched in the miCF secretome tended to be rather decreased in hypoxia-treated CF samples (Supplementary Fig. 7C).

In a final set of experiments, we made an attempt to validate our in-vitro data with miCF under in-vivo conditions. To this end, we specifically labeled proteins that were released via the ER secretory pathway by activated POSTN$^+$ CF, using the ER-localized biotin ligase ER-TurboID expressed under control of the POSTN promotor (workflow in Supplementary Fig. 8). An AAV9 vector system (AAV9-POSTN-ER-TurboID) was used for delivery. After i.v. injection of AAV9-POSTN-ER-TurboID at day 1 after MI (50 min ischemia followed by reperfusion) and biotin administration via the drinking water, hearts were perfused at day 5 post-MI in the Langendorff mode to capture the secreted biotinylated proteins in the coronary effluent, largely devoid of plasma proteins. Heart perfusion was necessary because POSTN is also expressed in the liver[32], resulting in ER-TurboID expression and thus biotinylation of liver-derived plasma proteins. As shown in Fig. 7A, we have identified 157 secreted proteins in the coronary effluent (Supplementary Data 4), of which 27 overlapped with the secretome proteins of cultured miCF (Supplementary Table 4). Comparing the effluent proteins of mice treated with AAV9-POSTN-ER-TurboID and of non-transduced control mice 5 days post-MI revealed 73 proteins to be significantly enriched (Fig. 7B), very likely representing effluent proteins originating from the POSTN$^+$ CF. Of these significantly enriched proteins, ten proteins overlapped with the miCF secretome (Fig. 7A, Supplementary Table 4) and eight of the ten proteins were preferentially expressed in miCF in comparison to other cell types present in the infarcted heart (Fig. 7C).

## Discussion

This study provides the first comprehensive overview of the secretome of CF in the infarcted heart as compared to the unstressed heart. It demonstrates the fibroblast-specific secretion of numerous paracrine/autocrine factors that can signal to surrounding cardiac cells to potentially influence cardiac remodeling. The newly identified proteins, verified by scRNAseq/snRNA-seq in mouse and human tissue samples, may serve as a rich source for novel therapeutic targets as well as biomarkers for CF activation in the context of injury-induced cardiac fibrosis.

A combination of several experimental advances made this study possible. Firstly, we minimized cell activation during the cell isolation procedure by applying a recently developed enzymatic technique which takes only 9 min for tissue digestion[17] and combined this with short-term culture (2 days) under serum-containing conditions. Secondly, we used click chemistry with AHA together with SILAC which permitted secretome analysis under optimal culture conditions. As a result, the majority (93%) of the identified proteins were predicted to be secreted (Fig. 2A). Finally, cell culture data were validated by specific in-vivo labeling of infarct-activated CF proteins via biotin ligase ER-TurboID expressed under control of the POSTN promotor. Together, it appears likely that results obtained under the chosen in-vitro conditions can be extrapolated to the in-vivo situation. This conclusion is further supported by reported scRNAseq/snRNAseq data of mice and humans, showing that most of the genes encoding identified proteins were rather selectively expressed in CF in comparison to other cardiac cell types.

A feature of the present secretome study is that miCF of the infarcted heart were collected from both the viable and infarcted areas. That was due to technical reasons: the desirable rapid (9 min) isolation of CF required a collagen perfusion method[17] which does not permit to differentiate any longer between infarcted and viable remote myocardium. As the roles of CF residing the remote zone and those located in the ischemic zone and there secretome are likely to be different, we made use of a recently published comprehensive spatial multi-omic data set of human myocardial infarction that includes transcriptomes of cells sampled from both zones at different time points post-MI[33]. Analysis of the spatial expression data of the proteins found to be significantly upregulated in the miCF secretome revealed that a group of proteins including COL1A1, LOX, and POSTN were higher expressed in CF located in the ischemic zone (Supplementary Fig. 9A). Interestingly, another group of proteins containing FMOD, GDF6, and SLIT2 were prevalently expressed by CF residing in the remote zone of infarcted hearts (Supplementary Fig. 9B). Together these data indicate that CF from both the ischemic zone and the remote zone of the infarcted hearts contributed to the significantly enriched proteins in the miCF secretome. This implies that CF in the non-ischemic remote zone also react to the MI, possibly triggered by alterations in the mechanical strain and/or soluble signals (local or circulating).

Under homeostatic conditions, the ECM composition is tightly controlled by the equilibrium between synthesis and degradation of its individual components. In line with this concept, we observed that in the secretome of unstressed cCF, collagens accounted for the largest fraction (24%) of ECM-associated proteins, followed by proteins assigned to matrix turnover as the second largest fraction (16%) (Fig. 2C). A more direct estimate of the turnover of individual secreted proteins was derived by relating the measured rate of secretion (secretome) to the cellular protein content (proteome) and transcription level (transcriptome) of each protein (Fig. 3). Of the proteins secreted under basal conditions (122), the respective intracellular protein was only measurable for a small number of proteins (32). Interestingly, the highest cellular pool sizes were found for FN1, HSP47 (Serpinh1), and collagen chains (COL1A1, COL1A2, COL3A1). Note, that among the collagen chains the highest intracellular levels were found for COL1A1, a major protein of the extracellular matrix (ECM), the turnover of which was estimated to be of the order of years[34]. Very likely it is the imbalance between accumulation of secreted proteins and their rate of degradation/tissue washout, that finally determines the biologically active concentration and leads to tissue fibrosis and cardiac dysfunction.

Of the paracrine/autocrine factors secreted by cCF, PAI-1 comprises with 68% the largest fraction (Fig. 2D). PAI-1, a serine protease inhibitor primarily known for regulation of fibrinolysis, is now additionally appreciated to function in many physiological processes including inflammation, wound healing and cell adhesion[35]. Interestingly, global PAI-1 knockout mice develop age-dependent cardiac-selective fibrosis and a similar phenotype was observed in a cardiomyocyte-specific PAI-1 knockout[36]. Thus, both cardiomyocyte- and CF-derived PAI-1 are likely to contribute to cardiac PAI-1 formation and thus may play a role in the preservation of cardiac performance under stress.

Analysis of the cellular distribution of gene expression of secreted proteins in reported scRNAseq/snRNAseq data revealed, that a large fraction of secreted ECM proteins was preferentially expressed in cCF

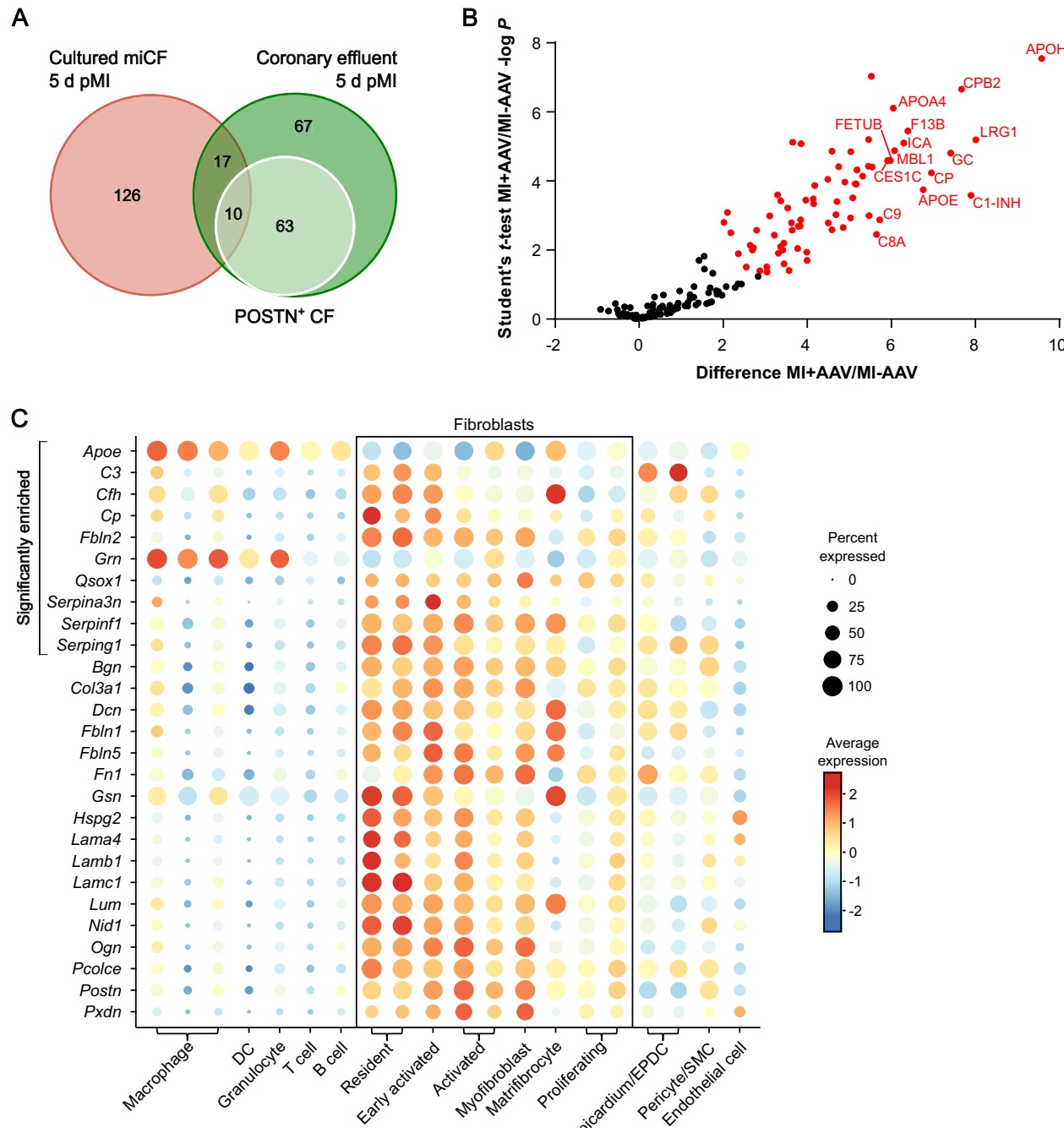

**Fig. 7 | In-vivo-secretome analysis of POSTN$^+$ CF 5 days post-MI.** Mice were transduced with AAV9-POSTN-ER-TurboID to allow biotin-labeling and subsequent enrichment of proteins secreted from POSTN$^+$ CFs in the coronary effluent (work flow in Supplementary Fig. 8). LC-MS/MS analysis identified 157 proteins in the coronary effluent 5 days after I/R surgery ($n = 6$; source data in Supplementary Data 4). These data were compared to the secretome data (153 proteins) of miCF isolated from hearts 5 days after MI ($n = 4$; source data in Supplementary Data 1). **A** Venn diagram[65] of identified proteins. Green circle, 157 proteins identified in the coronary effluent 5 days post-MI. White circle, 73 proteins significantly enriched in the effluent of AAV9-POSTN-ER-TurboID-transduced mice ($n = 6$) in comparison to non-transduced control mice 5 days post-MI ($n = 6$), representing the in-vivo-secretome of POSTN$^+$ CF. Red circle, 153 proteins in the miCF secretome 5 days post-MI. **B** Volcano plot of

proteins identified in the coronary effluent. Proteins with significantly different intensities in samples from TurboID-transduced mice (MI + AAV) and non-transduced control mice (MI-AAV) (Student's $t$-test-based SAM analysis, 5% FDR, $S_0 = 0.6$) are highlighted in red (73 proteins). Names of top 15 proteins with highest difference between MI + AAV and MI-AAV are annotated. The shown difference refers to the difference of group mean values of log$_2$ transformed intensities. **C** Dot plot visualizing transcript levels of the proteins identified in both data sets in the transcriptome of cell populations of $n = 3$ infarcted mouse hearts 5 days after I/R: proteins that were significantly enriched in the effluent of AAV9-POSTN-ER-TurboID-transduced mice (10 proteins) and proteins that did not reach significance in comparison to non-transduced mice (17 proteins). Previously published scRNAseq data of murine stromal and immune cell populations[29] were re-analyzed.

(Supplementary Fig. 3). This included various collagen chains, MMP2, fibulin 2 (FBLN2, reported to be essential for angiotensin II-induced myocardial fibrosis[37]), DCN (a proteoglycan possessing powerful anti-fibrotic, anti-inflammatory, antioxidant, and antiangiogenic properties[38]),

fibrillin 1 (FBN1, a constituent of the myocardial ECM in fibrosis[39]), and lumican (LUM, playing an important role in the cardiac fibrillogenesis[40]).

Among the paracrine/autocrine factors, the expression of GSN, PEDF, follistatin-related protein 1 (FSTL1), and AXL was predominantly found in

cCF (Supplementary Fig. 4). GSN has been reported to be an important mediator of angiotensin II-induced activation of CF and fibrosis[41], FSTL1 was shown to induce cardiac angiogenesis[42], AXL is likely to play a role in target organ inflammation[43], and PEDF was shown to display a protective role on endothelial tight junctions and the vascular barrier in the MI heart[44]. Together these data demonstrate, that non-activated CF secrete numerous biological active proteins which according to the literature are essential for balancing important biological processes such as angiogenesis, inflammation, and fibrosis in the unstressed heart.

In the secretome of infarct-activated miCF, we identified a total of 28 proteins to be significantly enriched. Besides 17 ECM-associated proteins, this included 11 paracrine factors (Fig. 4C), of which nine have a reported cardiovascular functionality (Table 1), such as promotion of thrombo-inflammation (semaphorin 7A, SEMA7A), signaling to cardiomyocytes (CPE, SLIT2, activin A), enhancement of angiogenesis (GDF6), and reduction of fibrosis (SFRP1). For several of the significantly enriched paracrine factors, for example SEMA7A, CPE, IGFBP2, and OPG, CFs have so far not been identified as a major site of production. Intriguingly, most of the secreted proteins were rather selectively expressed by the different CF populations in the heart in comparison to the other cardiac cell types (Fig. 5A, B), again supporting the specificity of our experimental approach.

As to the stimuli driving changes in the CF secretome post-MI, we tested TGF-β signaling and exposure to hypoxia as known specific stressors associated with myocardial ischemic injury. While in-vitro activation of TGF-β signaling in CF significantly altered the secretion of 22 proteins (Supplementary Fig. 7A), we did not observe significant changes after exposure to hypoxia (Supplementary Data 1). The lack of a secretory response to in-vitro hypoxia might be due to the adaption of CF to low oxygen conditions as was recently reported[45]. TGF-β stimulation on the other hand resulted in the enrichment of proteins that were also upregulated in the miCF secretome (LOX, FN1, INHBA, POSTN, COL1A1; Supplementary Fig. 7B). This suggests that TGF-β signaling very likely contributed to the shift from the cCF secretome to the miCF secretome. Which additional stimuli or combinations of conditions, such as inflammatory cytokines, mechanical stress, or other signaling pathways are involved in the modulation of the secretory process during ischemia needs to be elucidated in future studies.

The numerous proteins that we found to be secreted by CF 3 and 5 days after MI clearly indicates that CF activation and protein secretion is organized in a temporal fashion. This is in line with the view that fibroblasts are a dynamic cell type that achieves morphologically well-defined differentiated states following acute ischemic injury[30]. To decipher the functionality of individual factors within this secretome network, however, will not be easy and will constitute a major future task in the fibroblast field. Advanced cell culture techniques, such as organoids and in particular the heart-on-a-chip platform[46], appear to be promising and may lead the way for the future understanding of the intercellular signaling events.

The list of infarct-induced changes in the CF secretome may hold some promise for mitigating some of the deleterious consequences of MI. Intriguingly, many of the secreted proteins identified in mice CF were confirmed at the gene expression level in mice as well as in human CF in samples of acute MI (Fig. 5). Future therapeutic approaches may utilize the known anti-fibrotic activity of FMOD[47] and SFRP1[48] or pharmacologically neutralize the pro-fibrotic activity of BMP1, LOXL3, ECM1, and FN1 (Table 1). For some of the identified secreted proteins there is presently no cardiac function known; this includes DnaJ heat shock protein family (Hsp40) member B11, which was only detected in the miCF secretome.

The secreted factors also hold promise to serve as biomarkers for CF activation in the injured heart. IGFBP2, found in the present study to be significantly elevated in the miCF secretome (Table 1), was already reported as prognostic biomarker for heart failure[49]. Similarly, OPG (*Tnfrsf11b*), a cytokine belonging to the tumor necrosis factor receptor family which was significantly enhanced in the miCF secretome (Table 1), was also shown to

be elevated during the acute phase after MI[50] and to predict adverse cardiovascular events in stable coronary artery disease (PEACE trial)[51]. Comparing the expression level of secreted proteins between mice and human (Fig. 5A, B), the following proteins may be useful plasma marker in humans for the activated CF: COL1A1, COL12A1, COL16A1, FN1, LAMA2, LOX, LTBP2, PXDN, CRLF1, and SLIT2.

Communication of CF with cardiomyocytes was already suspected on the basis of single-cell analysis that identified CF as key constituent in the microenvironment to promote cardiomyocyte maturation[52]. The wide distribution of receptor expression indicates multiple potential interactions of CF secretome proteins with all cardiac cell populations (Supplementary Fig. 2). The interplay between CF and monocytes has already been reported to be of pivotal importance in the diabetic heart[53]. Along this line, in co-culture experiments the conditioned medium from cardiac mesenchymal stromal cells suppressed apoptosis of iPS-derived human cardiomyocytes[54]. Similarly, we found that the supernatant from miCF effectively attenuated apoptosis in hypoxia-stressed murine cardiomyocytes when cCF-conditioned medium served as control (Supplementary Fig. 5). In support of this finding, we identified in this study several proteins with reported cardioprotective activity, such as carboxypeptidase E (CPE), SFRP1, and SLIT2 (Table 1). Together these findings strongly suggest that MI-stressed CF secrete cardioprotective factors that play a quantitatively significant role in modulating I/R injury in cardiomyocytes. The individual contribution of the identified proteins to the cardioprotective effect needs to be elucidated in future studies.

Communication between cardiac cells appears to be bidirectional and importantly involves cross-talk within the cardiac microenvironment[55]. The results of our study shed light on the role of CF as signal source in the orchestration of cardiac homeostasis and post-injury remodeling. In the case of cCF, secreted proteins include PCSK6, IGFBP4, and NGF, which have been reported to inhibit cardiomyocyte senescence, act as cardiogenic growth factor, and exert cardioprotective activity, respectively (Supplementary Table 1). In the case of miCF, the pattern shifts and—as was discussed above—includes cardioprotective factors such as CPE, SFRP1, and Slit2 (Table 1). Recent studies have also demonstrated that CF forms functional electrical connections with cardiomyocytes[56]. It is therefore quite possible that CF more broadly influences cardiomyocyte function.

In the present study, we have explored the possibility to directly measure the CF secretome under in vivo conditions by using a vector encoding ER-localized biotin ligase ER-TurboID under control of the POSTN promotor to restrict expression to activated CF (AAV9-POSTN-ER-TurboID). However, this approach showed only limited overlap with the in vitro miCF secretome data. Firstly, POSTN, despite being specific for activated CF in the heart, is also strongly expressed in the liver[32], so that CF-derived biotinylated proteins are "contaminated", which we mitigated in our study by collecting the perfusate of the isolated heart. Secondly, because the number of CF within the heart is comparatively small, the intensity distribution of secreted biotinylated proteins from labeled precursor cells was only low. Thirdly, this in vivo approach was specifically targeted to POSTN+ CF populations within the infarcted heart[8], while enzymatic isolation of CF used for generation of miCF secretome data comprised all CF populations, including non-activated CFs of the remote regions. Thus, despite having shown the general feasibility of our labeling approach, future studies need to consider use of different CF promotors[57] or use of the advantages of TurboID in a transgenic approach[58].

In summary, this study is the first to report the secretome atlas for unstimulated (cCF) and infarct-activated fibroblasts (miCF). This secretome atlas contains aside of well-known extracellular matrix proteins numerous autocrine/paracrine proteins, of which several have previously not been assigned to CF. Most of the identified proteins are predominantly expressed in CF and a similar expression pattern was observed in human samples of AMI. Several of the identified proteins are likely to be critically involved in adverse cardiac remodeling and may serve as a rich resource for future diagnostic and therapeutic studies.

## Methods

### Animals

We have complied with all relevant ethical regulations for animal use. All experiments were performed according to the institutional and national guidelines for animal care in conformity with Directive 2010/63/EU and were approved by the Landesamt für Natur-, Umwelt- und Verbraucherschutz (reference number 81-02.04.2020.A351).

Male C57Bl/6J wildtype mice were purchased from Janvier (Le Genest Isle, France) and were used at an age of 10–12 weeks and body weight of 20–25 g. Animals were housed at the Central Facility for Animal Research and Scientific Animal Welfare (ZETT) of the Heinrich Heine University Düsseldorf, Germany, under standardized conditions.

MI by ischemia/reperfusion (I/R) was induced as described previously[59]. The left anterior descending coronary artery (LAD) was ligated with a 7-0 polypropylene thread for 50 min. Ischemia was checked by ST-segment elevation on the ECG. After 50 min of ligation, the polypropylene thread was released to initiate reperfusion. For sham surgery the same procedure was used, but without LAD ligation. Animals were given buprenorphine to manage post-operative pain. Severity assessment was performed daily (in the first 3 days after surgery, three times a day) by monitoring parameters including body weight, body condition, respiration, healing of the surgical scar, and behavior. Reaching the maximal severity score in one of the parameters or reaching the threshold in total scores was defined as a humane endpoint. No randomization or blinding was applied, and no exclusion criteria were set for CF isolation or coronary effluent collection (see below).

### CF isolation and cultivation

CF were isolated 5 days after MI or sham surgery as described previously[17]. Mice were sacrificed by cervical dislocation and the heart was cannulated via the aorta and perfused retrogradely at 37 °C with pre-warmed PBS for 3 min (constant flow of 2 ml/min) to wash out the residual blood from the coronaries. After washing, the system was switched to pre-warmed collagenase type II (Worthington Biochemical Corporation, Lakewood, USA) in PBS (1000 U/ml) for 8–9 min of digestion. To remove epicardial cells, hearts were bathed and shaken in collagenase solution which was discarded afterwards. Digested tissue was resuspended in cell culture medium (DMEM high glucose (P04-01597, PAN Biotech, Aidenbach, Germany), supplemented with 10% dialyzed FBS (Thermo Fisher Scientific, Waltham, USA) and 1% penicillin/streptomycin/glutamine (Merck Millipore, Burlington, USA)). The cell suspension was filtered using a 100 µl cell strainer and centrifuged for 1 min at $55 \times g$ at 15 °C to pellet cardiomyocytes. The supernatant was passed through a 40 µm cell strainer and centrifuged for 7 min at $300 \times g$. Resuspended cells were depleted for CD31$^+$ and CD45$^+$ cells (endothelial cells and immune cells, respectively) via magnetic depletion with Mojosort nanobeads and a Mojosort magnet (BioLegend, San Diego, USA) according to the manufacturer's protocol and seeded in cell culture medium. On the following day, cell debris was removed by stringent washing with PBS.

### CF cell viability assay

To evaluate cell viability during amino acid withdrawal and isotype labeling steps of the secretome analysis protocol, Cell Counting Kit 8 (Tebu-bio Le Perray-en-Yvelines, France), containing light-yellow tetrazolium salt that is reduced to an orange formazan by living cells, was used according to the manufacturer's protocol. CF were isolated from healthy mouse hearts as described above and $5 \times 10^3$ cells (in 100 µl) per well were seeded into a 96-well plate. On the next day, cells were washed with PBS and incubated in cell culture medium (see above) or depletion medium (DMEM high glucose customized without L-methionine (Met), L-arginine (Arg), and L-lysine (Lys) (PAN Biotech, Aidenbach, Germany), supplemented with 10% dialyzed FBS and 1% penicillin/streptomycin/glutamine) for up to 90 min. Subsequently, cells were washed with PBS, incubated in cell culture medium with CCK8 for 3 h, and absorbance was measured at 450 nm. To assess the effects of isotype labeling, cells were seeded as described above and

incubated in depletion medium for 60 min, followed by incubation in depletion medium supplemented with 0.1 mM azidohomoalanine (AHA) (Thermo Fisher Scientific), intermediate isotypes [$^{13}$C$_6$] Arg (84 µg/ml) and [4,4,5,5-D$_4$] Lys (146 µg/ml), heavy isotypes [$^{13}$C$_6$,$^{15}$N$_4$] Arg (84 µg/ml) and 146 µg/ml [$^{13}$C$_6$,$^{15}$N$_2$] Lys (Cambridge Isotope Laboratories, Tewksbury, USA), or combinations of AHA with intermediate and heavy isotypes for up to 24 h. CCK8 was added 3 h prior measurement. Statistical analysis was performed by Two-way ANOVA followed by Dunnett's multiple comparisons test using GraphPad Prism 9. As shown in Supplementary Fig. 1, Arg/Lys/Met withdrawal up to 90 min had no effect on CF cell viability (Supplementary Fig. 1A, Supplementary Data 5), while supplementation with AHA resulted in a significant reduction of CF cell viability after 16–24 h (Supplementary Fig. 1B, Supplementary Data 5). Thus, incubation times of 60 min for amino acid withdrawal and 8 h for isotype labeling were chosen for the secretome analysis protocol.

### Secretome analysis in cultured CF

For secretome analysis in short-term cultured CF by liquid chromatography-tandem mass spectrometry (LC-MS/MS), a modified protocol for robust protein quantification via SILAC labeling and click chemistry according to Eichelbaum and Krijgsveld[18] was used (see Fig. 1). To reach sufficient cell numbers without expansion in cell culture, CF preparations of three mouse hearts (3 and 5 days after sham or I/R surgery, respectively) were pooled for each sample and seeded into a 10 cm cell culture dish. After cultivation in cell culture medium for 24 h at 37 °C, cells were washed six times with PBS and incubated in Arg/Lys/Met depletion medium (4 ml/dish) for 1 h. Subsequently, cells were incubated in depletion medium (4 ml/dish) supplemented with 0.1 mM of the Met analog AHA and either intermediate or heavy Arg and Lys isotypes ([$^{13}$C$_6$] Arg (84 µg/ml), [4,4,5,5-D$_4$] Lys (146 µg/ml) or ([$^{13}$C$_6$, $^{15}$N$_4$) Arg (84 µg/ml), [$^{13}$C$_6$, $^{15}$N$_2$] Lys (146 µg/ml), respectively) for 8 h. Of the altogether four samples of each condition (sham /I/R surgery) at the individual time points, two were labeled with intermediate and two with heavy isotypes to exclude potential bias by the kind of labeling.

For in vitro stimulation with TGF-β, CF were isolated from healthy mouse hearts and seeded as described above. CF dishes were either treated with 10 ng/ml recombinant mouse TGF-β1 (bio-techne, Minneapolis, USA) or with 10 µM TGF-β receptor inhibitor SB-431542 (MedChemExpress, Monmouth Junction, New Jersey, USA) for 48 h. During the last 9 h of incubation, protein labeling was performed as described above by consecutive incubation with depletion medium (1 h) and medium supplemented with either intermediate or heavy Arg and Lys isotypes (8 h), in presence of TGF-β1 and SB-431542, respectively.

For in-vitro exposure to hypoxia, CF isolated from healthy mouse hearts and seeded as above were incubated in Arg/Lys/Met depletion medium for 1 h. After the medium was changed to medium supplemented with either intermediate or heavy Arg and Lys isotypes (see above), dishes remained in the standard $CO_2$ incubator (control samples) or were exposed to hypoxic conditions (1% $O_2$, 5% $CO_2$) in a Heracell 150i $CO_2$ incubator with oxygen control (Thermo Fisher Scientific) and incubated for 8 h.

After the labeling procedure, supernatants were collected and centrifuged for 5 min at $1000 \times g$ at 4 °C to remove cell debris. To stabilize proteins, 1X Halt Protease-Inhibitor-Cocktail (Thermo Fisher Scientific) was added and supernatants were stored at −80 °C until further use. For additional analysis of the intracellular proteome, remaining cells were washed four times with ice-cold PBS, scraped off in 1 ml PBS, and transferred to an 1.5 ml reaction tube. Cells were centrifuged for 5 min at $800 \times g$ at 4 °C and pellets were stored at −80 °C until further use.

For enrichment of newly synthesized proteins in the supernatants, intermediate- and heavy-labeled supernatants of sham/MI sample pairs were combined and concentrated using Pierce 3 kDA protein concentrators (Thermo Fisher Scientific) at 4000 x $g$ and 4 °C to a volume of about 250 µl. The Click-iT Protein Enrichment Kit (Thermo Fisher Scientific) was used according to the protocol of Eichelbaum and Krijgsveld[19] with slight modifications. In brief, 250 µl urea buffer, 100 µl of washed agarose resin,

10 µl of dissolved Reaction Additive 2, 1 µl 10 mM Copper (II) sulfate solution, and 63 µl Reaction Additive 1 were added to the samples and the samples were vortexed. To allow binding of the newly synthesized, AHA-labeled proteins to the agarose resin, samples were rotated overnight at 20 rpm at room temperature. On the following day, the supernatants containing non-bound proteins were removed by centrifugation for 1 min at $1000 \times g$ and the resin was washed four times with 900 µl 18 MΩ $H_2O$. To reduce the resin-bound proteins, the resin was resuspended in 500 µl SDS washing buffer (warmed to 37 °C) and 5 µl 1 M dithiothreitol. Samples were vortexed and heated for 15 min at 70 °C. After 15 min of cooling and centrifugation for 5 min at $1000 \times g$, the resin-bound proteins were alkylated by resuspending the samples in 500 µl freshly prepared 40 mM iodoacetamide solution. After shaking the samples at 500° rpm for 30 min in the dark, the resin was transferred to a Mini Bio-Spin Chromatography column (Bio-Rad Laboratories, Hercules, USA), which was placed in a 2 ml reaction tube, and stringently washed by centrifugation for 30 s at $1000 \times g$: 10 times with 0.8 µl of SDS washing buffer, 20 times with 0.8 µl 8 M urea buffer, 20 times with 0.8 µl isopropanol, and 20 times with 0.8 µl acetonitrile.

To digest the resin-bound proteins, samples were washed twice with 500 µl digestion buffer (100 mM Tris-HCl pH 8, 10% acetonitrile, 2 mM $Ca_2Cl$), each time transferring the samples to a new tube. The resin was pelleted for 5 min at $1000 \times g$ and supernatants was discarded, leaving about 200 µl of resin in the tube. After addition of 0.5 µg trypsin in a volume of 0.5 µl, samples were briefly vortexed and then shaken overnight at 500 *rpm* at 37 °C. On the next day, the resin was pelleted for 5 min at $1000 \times g$ and the peptide-containing supernatants were transferred to a new tube. The resin was resuspended in 500 µl 18MΩ $H_2O$ and again centrifuged for 5 min at $1000 \times g$. The resulting supernatants were added to the supernatants of the step before and the combined volume was filled up with 18 MΩ $H_2O$ to a total volume of 1 ml to dilute the acetonitrile. To acidify the samples, 20 µl of 10% trifluoroacetic acid (TFA) was added. For desalting, Sep-Pak Cartridges (Vac 1cc (50 mg) tC18, Waters, Milford, USA) were prepared by placing them in a 20-position cartridge that was connected to a vacuum pump and washing with 900 µl acetonitrile, 300 µl of 50% acetonitrile, and 0.5% acetic acid. After equilibrating the cartridges with 900 µl 0.1% TFA, the samples were loaded onto the cartridges, washed with 900 µl 0.1% TFA and 900 µl 0.5% acetic acid, and eluted with 300 µl 50% acetonitrile and 0.5% acetic acid. Samples were dried using a rotational vacuum concentrator (Christ, Osterode am Harz, Germany) and stored at −80 °C until further use.

Lysates of frozen CF cell pellets (see above) were prepared as described earlier[60]. Briefly, frozen CF were lysed by bead-milling in $2 \times 30$ µl cell lysis buffer (30 mM Tris-HCl; 2 M thiourea; 7 M urea; 4% CHAPS (w/v) in water, pH 8.5). Cleared cell lysates of intermediate- and heavy-labeled CF of sham/MI sample pairs were combined (2.5 µg protein of each) and were shortly stacked into a polyacrylamide gel. Subsequently, the gel was stained with Coomassie Brilliant Blue and the protein-containing band was cut out, de-stained, reduced with dithiothreitol, alkylated with iodoacetamide, and digested with trypsin overnight. Resulting peptides were extracted from the gel, dried in a vacuum concentrato,r and 500 ng of peptides were prepared for mass spectrometric analysis in 0.1% trifluoroacetic acid.

Peptides generated from click chemistry-enriched CF supernatant proteins and CF cell lysates, respectively, were separated on C18 material on an Ultimate3000 rapid separation LC system (Thermo Fisher Scientific) using a 2 h gradient as described earlier[60].

Separated peptides were subsequently injected by an electrospray nano-source interface into a Fusion Lumos mass spectrometer (Thermo Fisher Scientific) operated in data-dependent, positive mode. First, survey scans were recorded in the orbitrap analyzer (resolution: 120,000, target value advanced gain control: 250,000, maximum injection time: 60 ms, scan range 200–2000 m/z, profile mode). Second, 2-7fold charged precursor ions were selected by the quadrupole of the instrument (isolation window 1.6 m/z), fragmented by higher energy collisional dissociation (collision energy 35%), and analyzed in the linear ion-trap of the instrument (scan rate: rapid, target value advanced gain control: 10,000, maximum injection time: 50 ms,

centroid mode). Cycle time was 2 s; already fragmented precursors were excluded from fragmentation for the next 60 s.

Data analysis, including peptide and protein identification and MS1-based quantification was carried out with MaxQuant version 2.1.3.0 (MaxQuant version 2.0.3.0 for in-vitro stimulation experiments with TGF-β or hypoxia; Max Planck Institute for Biochemistry, Planegg, Germany) separately for the different sample batches with standard parameters, if not stated otherwise. MaxQuant was chosen because in our hands, it shows a more reliable performance on SILAC datasets. For searches, a multiplicity of three was chosen (channel light: Arg+0 and Lys+0, channel intermediate: Arg+6, Lys+4, channel heavy: Arg+10, Lys+8) and the "match between runs" function enabled. Searches were carried out on the basis of 55341 *mus musculus* proteome sequences (UP000000589) downloaded from UniProt KB on 18th January 2022. Quantitative data was further processed by Perseus 1.6.6.0 (Max Planck Institute for Biochemistry, Planegg, Germany) and Excel. Here, proteins were filtered for at least two identified peptides. Intensities were normalized based on the median intensity per sample separately for the light and intermediate/heavy channels. It should be noted, that measured intensities of individual proteins can depend on available tryptic peptides, their properties, and modifications and therefore might slightly vary from the absolute protein amount. For the identification of differentially abundant proteins between CF from sham control and post-MI hearts, only proteins were considered that showed at least three valid values in one sample group. For the identification of differentially abundant proteins in CF after in vitro stimulation with TGF-β or hypoxia, at least four valid values had to be present in one sample group. Differences were determined using the significance analysis of microarrays method[61] based on Student's $t$ tests and permutation-based control of the false discovery rate (FDR) which is required to account for multiple testing (FDR = 5%; if not stated otherwise: lysate samples: $S_0 = 0.6$, secretome samples: $S_0 = 0.1$). Tests were calculated on $\log_2$-transformed normalized intensity values with missing values filled in with random values drawn from a downshifted normal distribution (downshift: 1.8 standard deviations, width: 0.3 standard deviations). Identified proteins were annotated using gene ontology categories by Perseus, and the secretion behavior of proteins identified were predicted by OutCyte[62]. Differences in mean values of $\log_2$-transformed normalized intensities were used for a one-dimensional annotation enrichment analysis[63] to identify functionally related proteins which were collectively altered in their abundance.

Data were visualized using GraphPad Prism 8 and Microsoft Excel as well as Circos[64] 0.69 (https://circos.ca/), DeepVenn[65] (https://www.deepvenn.com/), and Cytoscape[66] 3.9.1 (https://cytoscape.org/).

### In-vivo-secretome analysis in POSTN⁺ CF

For in-vivo-secretome analysis of CF by LC-MS/MS, a modified protocol for cell type-selective secretome profiling according to Wei et al.[67] was used (see Supplementary Fig. 8). This is based on proximity labeling of proteins passing through the ER secretory pathway by ER-located biotin ligase TurboID (ER-TurboID), which is expressed under control of a cell type-specific promotor.

Plasmid pAAV-TBG-ER-TurboID was purchased from Addgene (Watertown, USA) and the TBG promotor was replaced by the POSTN promotor identified by Lindsley et al.[68] to target activated CF expressing the activation marker POSTN. This pAAV-POSTN-ER-TurboID was transduced using adeno-associated virus serotype 9 (AAV9-POSTN-ER-TurboID) by injecting the virus particles ($1 \times 10^{12}$ genome copies) into the tail veins of mice at day 1 post-MI induction by I/R surgery. Mice with I/R surgery but without virus injection were used as negative control for ER-TurboID-mediated protein labeling. Biotin was provided via the drinking water (0.5 mg/ml) to both mouse groups at days 2, 3, and 4 post-MI. At day 5 post-MI, secreted cardiac proteins were harvested by collecting the coronary effluent during Langendorff-based retrograde perfusion. To this end, explanted hearts were cannulated via the aorta and perfused with oxygenated Krebs-Henseleit buffer under constant pressure (1 m $H_2O$). After 10 min initial perfusion to wash out residual blood, coronary effluent was

collected for 60 min in 50 ml centrifuge tubes placed on ice. 1X cOmplete protease inhibitor cocktail (1 tablet/50 ml; Roche, Basel, Switzerland) was added to the collected effluent samples to prevent protein degradation. Effluent samples (~45–60 ml) were concentrated to 250 µl using Pierce 3 kDa protein concentrators (Thermo Fisher Scientific) and stored at −80 °C until further use.

Enrichment of biotinylated proteins was performed according to protocols by Wei et al. [67] and Cheah and Yamada[69] using Dynabeads MyOne Streptavidin T1 (Thermo Fisher Scientific). A volume of 200 µl Dynabeads was washed twice with 1 ml lysis buffer (50 mM Tris HCl, 150 mM NaCl, 0.4% SDS, 1% NP-40, 1 mM EGTA, 1.5 mM $MgCl_2$, 1X cOmplete protease inhibitor cocktail (1 tablet/50 ml; Roche), pH 7.4) by centrifugation for 2 min at $2000 \times g$. The pellet was resuspended in 100 µl lysis buffer and 200 µl of the concentrated effluent was added. Samples were rotated overnight at 20 rpm and 4 °C. Samples were washed with 1 ml lysis buffer, 1 ml washing buffer (50 mM Tris HCl, 2% SDS, pH 7.5) and twice with 1 ml lysis buffer. For elution, samples were incubated for 5 min at 95 °C after addition of 15 µl of a 25 mM biotin solution. Samples were centrifuged for 2 min at $2000 \times g$ and supernatants containing biotinylated proteins were transferred into 2 ml reaction tubes and stored at −80 °C until further use.

For LC-MS/MS analysis, eluted proteins were shortly stacked into a polyacrylamide gel. After staining with Coomassie Brilliant Blue, the protein-containing bands were cut out, de-stained, reduced with dithiothreitol, alkylated with iodoacetamide and digested with trypsin overnight as described[60]. After extraction from the gel, the peptides were dried in a vacuum concentrator and 1/3rd of the peptides were prepared for mass spectrometric analysis in 0.1% trifluoroacetic acid.

Samples were analyzed on a Fusion Lumos mass spectrometer (Thermo Fisher Scientific) in data-dependent, positive mode, in a similar setting as described above for secreted proteins but with some modifications: the gradient length for peptide separation was only 1 h, and some settings for spectrum acquisition were different. Full scans were recorded in the orbitrap (resolution: 120,000, scan range: 200–2000 m/z, maximum injection time: 60 ms, automatic gain control target: 400,000, profile mode), 2–7 fold charged precursors selected for quadrupole-based isolation (1.6 m/z isolation window), fragmented by higher-energy collisional dissociation (collision energy 35%) and fragment spectra recorded in the linear ion trap of the instrument (scan rate: rapid, scan range: auto, maximum injection time: 150 ms, automatic gain control target: 10,000, centroid mode). Cycle time was 2 s and already isolated precursors were excluded from isolation for the next 60 s.

Data analysis was carried out with Proteome Discoverer 2.4.1.15 (Thermo Fisher Scientific) in order to profit from an enhanced sensitivity in comparison to MaxQuant-based searches. For protein identification, sequences as indicated above for cultured CF were used for protein identification by Sequest HT (10 ppm and 0.6 Da mass deviation for precursor and fragment masses, respectively; variable modification: methionine oxidation, N-terminal acetylation, and methionine loss; fixed modification: carbamidomethyl on cysteines). Proteins and peptides were accepted based on the Percolator node with a false discovery rate of 1%. The precursor quantification node was used for peptide and protein quantification. Only "high confidence" proteins were reported, which were identified with at least two different peptides and did not occur in the list of potential contaminant sequences. Statistical data analysis was carried out as described above for isolated CF samples, using an $S_0$ of 0.6 and 5% FDR on $\log_2$-transformed normalized intensities. To further increase confidence, proteins detected in <4 of the 6 samples in the virus-injected mouse group were excluded from further analysis.

### Immunofluorescence analysis
To analyze the distribution of ER-TurboID expression in the healthy and post-MI heart, AAV9-POSTN-ER-TurboID virus particles were injected into the tail vein ($1 \times 10^{12}$ genome copies) 2 days after I/R or sham surgery. Biotin was provided via the drinking water at days 4, 5, and 6 post-MI. At day 7 post-MI, mice were sacrificed, and hearts were embedded in tissue freezing medium. Cryostat sections were prepared and fixed with 4% paraformaldehyde (PFA) for 15 min. Immunostaining was performed using primary antibodies specific for POSTN (1:100; CAT# TA309212, OriGene, Rockville, USA) and V5 tag (1:100; clone 2F11F7, CAT# 37-7500, Thermo Fisher Scientific) to detect V5-tagged ER-TurboID, and secondary antibodies AF488-coupled goat-anti-rabbit IgG and AF594-coupled goat-anti-mouse IgG (1:1000; Thermo Fisher Scientific) in presence of 0.2% saponin. Cover slips were mounted with ProLong Gold Antifade Reagent with DAPI to label cell nuclei. Fluorescence microscopy was performed using a BX61 fluorescence microscope (Olympus, Tokyo, Japan). Images were processed for publication using ImageJ/Fiji[70].

### Cardiomyocyte cell protection assay
To explore the effects of factors secreted by post-MI CF on cardiomyocyte cell viability, medium either conditioned by CF from mouse hearts 5 days post-MI or CF from healthy mouse hearts as control was generated. CF were isolated as described above and seeded into 6-well cell culture plates (CF from 1 heart in 1 well) in cell culture medium with 20% FBS. After 24 h, cells were washed with PBS and fresh cell culture medium with 20% FBS was added (1 ml/well). After an additional 24 h cultivation, the conditioned medium was collected and stored at −80 °C.

For the isolation of murine primary adult cardiomyocytes, a healthy mouse heart was placed in ice-cold Tyrode buffer (in mM: NaCl 11.3, KCl 0.47, $KH_2PO_4$ 0.06, $Na_2HPO_4$ 0.06, $MgSO_4$ 0.12, $NaHCO_3$ 1.2, $KHCO_3$ 1, HEPES 1, taurine 3, (-)-blebbistatin 0.01, and glucose 5.5). The heart was cannulated via the aorta to facilitate retrograde perfusion with an enzyme solution containing 0.014% trypsin-EDTA (15090046, Gibco, Thermo Fisher Scientific) and 0.075 mg/ml Liberase (5401127001, Merck Millipore) for 5 min. After digestion, excess tissue (e.g., aorta, atria) was removed and discarded. The ventricles were transferred in a dish with 3 ml enzyme solution and pulled apart with fine forceps. The remaining tissue was gently triturated and the suspension filtered through a 200 µm polyamide sieve cloth (4-1413, neoLab, Heidelberg, Germany). The enzymatic digestion was quenched by addition of 3 ml Tyrode buffer containing 20% FBS. The cardiomyocytes were gravity-sedimented in round-bottom 14 ml-tubes for 10 min at room temperature. After sedimentation, the solution was aspirated and the pelleted cells were repeatedly gently resuspended and centrifuged in solutions with increasing calcium concentrations (0.1 mM calcium chloride, 0.2 mM calcium chloride, 0.4 mM calcium chloride, 0.8 mM calcium chloride). Lastly, the final of the four calcium solutions was completely and carefully aspirated and the cardiomyocytes were resuspended in plating medium (M199 medium containing 10% FBS, 1x antibiotic-antimycotic (15963194, Gibco, Thermo Fisher Scientific), 1x insulin-transferrin-selenium (12097549, Gibco, Thermo Fisher Scientific), 10 µM blebbistatin, 5 mM creatine, 2 mM L-carnitine, and 5 mM taurine).

Freshly isolated primary adult cardiomyocytes were seeded on laminin-coated wells (5000 cells/well) of two 96-well tissue culture plates and cultured for 1 h. Subsequently, one of the plates was exposed to hypoxia by exchanging the medium to ischemia buffer (in mM: NaCl 119, KCl 5.4, $MgSO_4$ 1.3, $NaH_2PO_4$ 1.2, HEPES 5, $MgCl_2$ 0.5, $CaCl_2$ 0.9, Na-lactate 20, blebbistatin 0.01, and BSA 0.1%; 1% $O_2$) and transferring the plate into a hypoxia chamber (1% $O_2$). As control, the second plate received fresh culture medium and remained in the cell culture incubator (5% $CO_2$ and 20% $O_2$). After 90 min incubation, the hypoxic plate was transferred back to the cell culture incubator and cardiomyocytes in both plates received medium conditioned by either CF from healthy hearts or post-MI hearts (see above), supplemented with blebbistatin (0.01 mM). After 18 h incubation, live/dead imaging was performed using 0.008% Trypan Blue. Eight field-of-view images in 100× magnification were captured in triplicate wells, and live/dead cardiomyocytes were quantified.

### Metabolic RNA sequencing
To quantify newly synthesized and existing RNA transcripts in CF, thiol(SH)-linked alkylation for the metabolic sequencing (SLAMseq) using the SLAMseq Explorer and Kinetics Kit (Lexogen, Vienna, Austria) was

performed. CF isolated from healthy hearts was seeded into a 24-well plate (30,000 cells/well, two wells/sample) and grown until confluency. Then, cells were cultured in culture medium with the uridine analog 4-Thiouridine (S4U, 0,1 mM) to allow S4U incorporation into newly synthesized RNA transcripts. As negative control, cells were incubated in parallel without the addition of S4U. After 12 h, CF were lysed by TRIsure (Meridian Bioscience, Cincinnati, USA). RNA isolation and alkylation by iodoacetamide, which results in incorporation of a guanine instead of an adenine at S4U nucleotides during downstream NGS library preparation, were performed according to the kit manufacturer's protocol, including the optional addition of unique molecular identifiers (UMI) for later deduplication.

The QuantSeq 3' mRNA-Seq Library Prep Kit FWD (Lexogen, Vienna, Austria) was used to prepare NGS libraries with the optional UMI added for duplicate removal in later stages of the analysis. At least 500 pg of alkylated RNA samples were used as input. High-throughput sequencing of the libraries was performed using the Illumina Next-Seq1000/2000 system (Illumina Inc., San Diego, USA) with a target of 50mio single-end, 75 bp long-reads per sample. Demultiplexing was performed using bcl2fastq2 as part of a snakemake-based pipeline (https://github.com/WestGermanGenomeCenter/bcl2fastq2_Pipeline). Sequence quality was checked using FastQC, MultiQC, and the SLAM-DUNK software[71] utilizing its uttrates function. T>C conversions, which result from the incorporation of guanine at alkylated S4U nucleotides during NGS library preparation, were detected and quantified with one minor addition. Briefly, the previously added UMI were moved from the reads sequence into the read header using umi-tools (https://umi-tools.readthedocs.io/en/latest/) extract function, discarding all reads without complete UMI and TATA-spacer sequence before mapping. After the mapping step of SLAM-DUNK (using mm39 as the reference genome), mapped reads were deduplicated with umi-tools dedup. Further analysis of the SLAM-DUNK output data was performed using R. Transcripts without T detection in a sample were removed. A beta-binomial test was performed according to manufacturer's instructions using the bb.test function. P values were calculated to detect transcripts with significantly increased T>C conversion rate ($P < 0.05$).

### Single-cell/single-nucleus RNA sequencing data analysis

Single-cell RNA sequencing (scRNAseq) and single-nucleus RNA sequencing (snRNAseq) data were processed using the R Seurat package[72] (v3.0).

For comparison of gene expression among cardiac cell types, including cardiomyocytes, an snRNAseq data set (whole heart sample) of cardiac cells isolated from a healthy heart of a 12-week-old mouse[23] (ArrayExpress E-MTAB-7869, sample Y1) was used.

For comparison of gene expression in CF, 5 days after sham and MI surgery or among the cardiac cell types 5 days post-MI, we re-analyzed scRNAseq data sets (whole heart samples) previously published by us[29]. Names of CF populations were annotated according to expression of CF population markers on the basis of Farbehi et al.[73] and Shi et al.[74].

To assess gene expression in human healthy hearts, post-MI hearts, and hearts with cardiomyopathy, we analyzed snRNAseq and cellular indexing of transcriptomes and epitopes by sequencing (CITE-seq) data sets recently generated[24–26]. Tissue samples for these data sets were acquired from the left ventricular apex (fresh for CITE-seq and frozen for snRNAseq) and included non-failing controls (rejected for heart transplant but preserved ejection fractions), acute myocardial infarction (within 3 months of MI), and chronic heart failure (ischemic and non-ischemic cardiomyopathy). Tissue from the left ventricular apex was used in all samples to provide consistency in comparing signatures from sample to sample.

To compare gene expression in human CF located in the post-MI heart either in the ischemic zone or the remote zone, we used snRNAseq data sets generated within a spatial multi-omic map of human MI[33] (https://cellxgene.cziscience.com/collections/8191c283-0816-424b-9b61-c3e1d6258a77). Data from ischemic zone and remote zone samples

2 days (patients no. P3 and P9) and 5 days (patient no. P2) post-MI were analyzed.

### Statistics and reproducibility

The methods used are described in detail in the respective 'Secretome analysis in cultured CF', 'In-vivo-secretome analysis in POSTN⁺ CF', and 'Metabolic RNA sequencing' sections and in the figure legends (where appropriate). P values of <0.05 were considered statistically significant. Experiments were performed with at least $n = 3$ biological replicates. For secretome analysis in cultured CF, experiments were performed with biological replicates in terms of CF preparations from different animals. For in-vivo-secretome analysis, experiments were performed with biological replicates in terms of coronary effluents collected from different animals. Numbers of biological replicates are stated in the figure legends where appropriate.

### Reporting summary

Further information on research design is available in the Nature Portfolio Reporting Summary linked to this article.

### Data availability

Mass spectrometry data have been deposited to the ProteomeXchange Consortium via the PRIDE[75] partner repository with the dataset identifiers PXD046238, PXD053791, and PXD060926. Metabolic RNA sequencing data have been deposited in NCBI's Gene Expression Omnibus[76] and are accessible through GEO Series accession number GSE293090. Source data underlying the presented graphs and charts are provided in the Supplementary Data files. All other data supporting the findings of this study are available from the corresponding authors upon reasonable request.

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

## Acknowledgements

This work was funded by the Deutsche Forschungsgemeinschaft (DFG, German Research Foundation)—458365199. This work was supported by the DFG Research Infrastructure West German Genome Center, project 407493903, as part of the Next Generation Sequencing Competence Network, project 423957469.

## Author contributions

J.B.: Conceptualization, methodology, validation, formal analysis, investigation, writing—original draft, writing—review & editing, visualization, project administration. G.P.: methodology, validation, formal analysis, investigation, resources, data curation, writing—original draft, writing—review & editing, visualization, project administration. A.J.: Methodology, resources. M.B.: Methodology, Resources. Z.D.: Investigation, resources. J.V.: Methodology, investigation. R.Z.: Investigation. J.St.: Investigation. A.A.E.M.: Investigation. T.W.: Methodology, formal analysis, investigation. D.R.: Methodology, formal analysis, investigation, data curation. T.L.: Formal analysis, investigation, resources, data curation. C.A.: Investigation, resources. J.A.A.: Formal analysis, resources, data curation, visualization. K.J.L.: Resources, supervision. K.K.: Methodology, resources, supervision. B.L.: Methodology, resources, supervision. P.M.: Methodology, resources, supervision. K.S.: Methodology, resources, supervision, writing—original draft. J.H.: Conceptualization, formal analysis, investigation, writing—original draft, writing—review & editing, visualization, project administration, funding acquisition. J.Sch.: Conceptualization, supervision, writing—original draft, writing—review & editing, project administration, funding acquisition.

## Funding

## Competing interests
The authors declare no competing interests.
