## [Transparent Peer Review file · Communications Biology]

A secretome atlas of cardiac fibroblasts from healthy and infarcted mouse hearts

Corresponding Author: Professor Jürgen Schrader

Version 0:

Reviewer comments:

Reviewer #2

(Remarks to the Author)

This is an interesting study of the cardiac fibroblast secretome using some novel techniques. The accompanied transcriptomic and intracellular proteomic analyses allowed the further exploration of the regulation of cardiac fibroblast protein secretion. Only a few minor limitations/issues were noticed.

1. One limitation of this study is the identification of secreted proteins only in the medium. Many secreted proteins, such as ECM proteins, are primarily attached to the cell culture plates. The levels of these proteins measured in this study were likely underestimated, which may explain the observation "Similarly, collagen chains (COL1A1, COL1A2, COL3A1) showed high intracellular protein pool sizes and high mRNA transcript levels, but comparatively little secretion (Figure 3)."
2. The accuracy of Figure 7A legend needs to be double-checked. "White circle, 27 proteins significantly enriched in the effluent of AAV9-POSTN-ER-TurboID-transduced mice (n=6) in comparison to non-transduced control mice 5 days post-MI (n=6), representing the in-vivo-secretome of POSTN+ CF." There are 73 but not 27 proteins in the white circle. It seems that only 73 of 157 identified proteins were secreted from CFs. The others were possibly background. 10 of the 27 proteins that are secreted by CFs both in vitro and in vivo belong to the 73 proteins. The analysis shown in Figure 7C should highlight the 10 proteins.

Reviewer #3

(Remarks to the Author)

In this study, the authors measured and defined the secretome atlas of cardiac fibroblasts in control and post-MI hearts. The authors aimed to link expression status and fibroblast functional role in the ischemic LV. Fibroblasts isolated from control hearts were compared to that at 3 and 5 days post cardiac injury to characterize cellular function. A detailed description of concerns are listed below.

- 1) The study performed does an excellent job combining proteomic analysis both in vitro and in vivo to demonstrate potential impacts of fibroblast post-MI however, much of what was done is observational and does not fully define how the secreted proteins would affect function.
- 2) There have been many studies that have evaluated fibroblast phenotypes over time post-MI using genomic techniques so the addition of the secreted profile to support genomic findings is a nice addition. However, it is unclear what new information this is bringing to the field. Additional studies demonstrating the effect of certain proteins secreted by the fibroblasts or what stimuli are causing the fibroblast to secrete the different proteins would greatly improve this manuscript.
- 3) Were the fibroblasts collected from both the viable and infarcted areas of the heart? As their role and what they secrete is likely vastly different, it would be interesting to stain the hearts for some of the key proteins identified to determine potential spatial differences.
- 4) I know that the human data was a previously reported dataset however additional details regarding the type of tissue collected (region of the heart, rejected transplant versus device placement collection, etc.) would be helpful seeing as part of the study is comparing the mouse and human findings. If they are from different, this would be a limitation that should be discussed.

Version 1:

Reviewer comments:

Reviewer #2

(Remarks to the Author)

The authors have appropriately addressed my comments.

Reviewer #3

(Remarks to the Author)

Authors addressed all concerns. No new comments to add.

Reviewer #1

This is an interesting study of the cardiac fibroblast secretome using some novel techniques. The accompanied transcriptomic and intracellular proteomic analyses allowed the further exploration of the regulation of cardiac fibroblast protein secretion. Only a few minor limitations/issues were noticed.

We would like to thank this reviewer for his careful reading of our manuscript and finding it generally interesting.

1. One limitation of this study is the identification of secreted proteins only in the medium. Many secreted proteins, such as ECM proteins, are primarily attached to the cell culture plates. The levels of these proteins measured in this study were likely underestimated, which may explain the observation “Similarly, collagen chains (COL1A1, COL1A2, COL3A1) showed high intracellular protein pool sizes and high mRNA transcript levels, but comparatively little secretion (Figure 3).”

You are raising an important point and we fully agree that some ECM proteins, particularly those with poor solubility, may have preferentially attached to the cell culture plates thereby escaping detection. This limitation of our secretome approach is now clearly addressed in the revised version of the manuscript (lines 135-137).

2. The accuracy of Figure 7A legend needs to be double-checked. “White circle, 27 proteins significantly enriched in the effluent of AAV9-POSTN-ER-TurboID-transduced mice (n=6) in comparison to non-transduced control mice 5 days post-MI (n=6), representing the in-vivo-secretome of POSTN+ CF.” There are 73 but not 27 proteins in the white circle. It seems that only 73 of 157 identified proteins were secreted from CFs. The others were possibly background. 10 of the 27 proteins that are secreted by CFs both in vitro and in vivo belong to the 73 proteins. The analysis shown in Figure 7C should highlight the 10 proteins.

Thank you for making us aware of the inaccuracy in Figure 7A. We apologize! We have made the appropriate corrections (line 1131) and again carefully checked for accurate data representation. The protein/gene order in Figure 7C has been re-arranged, now allowing easier identification of the 10 proteins, that were detected both in the in-vitro secretome analyses and were also significantly enriched in samples from AAV9-POSTN-ER-TurboID-transduced mice. The corresponding text passage in the Result section has been modified to more clearly focus on these proteins (lines 275-278).

Finally we would like to add that, initiated by the comments of **Reviewer #2**, the revised version of the manuscript now contains additional new experimental data addressing the following issues:

1. We have included functional data showing that the supernatant of infarct-activated miCF displays anti-apoptotic activity on primary mouse cardiomyocytes which were subjected to hypoxia in an in-vitro assay simulating ischemia/reperfusion injury (Supplementary Figure 5).
2. We studied in-vitro the influence TGF- β treatment and exposure to hypoxia on the secretome of CF using the same technology as elaborated for cCF and miCF. We found that TGF- β induced the release of some of the proteins that we observed to be enriched in the miCF secretome

(Supplementary Figure 7B). In contrast, hypoxia (1% O₂ versus 20% O₂) did not induce a significant change in the release of proteins (Supplementary Data 1).

3. We have included spatial expression data for selected miCF proteins derived from a published data set of the human post-MI-heart, indicating that a group of the proteins are higher expressed in fibroblasts located within the ischemic zone (e.g. POSTN, LOX, COL1A1; Supplementary Figure 9A), as one might have expected. Interestingly, some proteins (e.g. FMOD, GDF6, SLIT2; Supplementary Figure 9B) were prevalently expressed by fibroblasts located in the remote zone. This suggests that in response to cardiac ischemia, also the fibroblasts in the adjacent non-ischemic remote areas respond. This might be due to alterations in the mechanical strain and/or soluble signals (local or circulating).

Reviewer #2:

In this study, the authors measured and defined the secretome atlas of cardiac fibroblasts in control and post-MI hearts. The authors aimed to link expression status and fibroblast functional role in the ischemic LV. Fibroblasts isolated from control hearts were compared to that at 3 and 5 days post cardiac injury to characterize cellular function. A detailed description of concerns are listed below.

1) The study performed does an excellent job combining proteomic analysis both in vitro and in vivo to demonstrate potential impacts of fibroblast post-MI however, much of what was done is observational and does not fully define how the secreted proteins would affect function.

Thank you for your thoughtful and constructive feedback. We also greatly appreciate your recognition of having successfully integrated both in-vitro and in-vivo proteomic analyses.

Defining how secreted proteins influence function is a critical yet complex task, and we have given considerable thought to this challenge. As you know, cardiac fibroblasts are not a homogeneous population, and our current understanding of the specific cellular locations of each subpopulation remains limited. Furthermore, we have identified a range of biologically active proteins that are secreted by these cells, which, according to existing literature, exhibit diverse functionalities, including pro- and antifibrotic, angiogenic, anti-apoptotic, and thrombo-inflammatory effects (see Table 1). We believe that a comprehensive understanding of these distinct functions requires advanced methodologies, such as Organs-on-Chips (OoCs), which can better replicate human physiology and allow for a more nuanced analysis. This, however, is beyond the scope of this manuscript.

Nevertheless, we have done additional experiments (in cooperation with Jens Vogt and Bodo Levkau, who are new coauthors) and have incorporated new functional data that highlight a significant cardioprotective activity in supernatants of CF isolated from infarcted hearts. Specifically, we assessed anti-apoptotic activity by treating isolated primary mouse cardiomyocytes, subjected to hypoxia to simulate ischemia/reperfusion (I/R) injury, with supernatants derived from fibroblasts isolated from both healthy control hearts (cCF) and MI-injured hearts (miCF). Our results show that supernatant from MI-derived miCF effectively attenuated apoptosis in hypoxia-stressed cardiomyocytes (now shown in Supplementary Figure 5). These findings strongly suggest that MI-stressed miCF secrete cardioprotective factors that play a quantitatively significant role in modulating I/R injury in cardiomyocytes. In support of this notion, we found in this study several proteins with reported cardioprotective activity, such as carboxypeptidase E (CPE), Wnt/ β -catenin pathway inhibitor SFRP1, and SLIT2, to be secreted by miCF (see Table 1). The results of these additional experiments have been included in the revised manuscript (lines 193-201), along with a more detailed discussion of the anti-apoptotic cardioprotective functionality (lines 408-427).

2) There have been many studies that have evaluated fibroblast phenotypes over time post-MI using genomic techniques so the addition of the secreted profile to support genomic findings is a nice addition. However, it is unclear what new information this is bringing to the field. Additional studies demonstrating the effect of certain proteins secreted by the fibroblasts or what stimuli are causing the fibroblast to secrete the different proteins would greatly improve this manuscript.

In our view, the new information our study is bringing to the field is that it is the first secretome study of cardiac fibroblasts performed with advanced proteomic techniques together with rapid cell isolation

techniques to capture the in-vivo situation. We found many known matrix proteins but also quite a number of new paracrine factors. We therefore consider our study to be valuable as a resource for the cardiovascular research community.

Your question regarding the stimuli that trigger cardiac fibroblasts to secrete different proteins is an important one. To address this issue, we conducted additional experiments to investigate the effects of specific stimuli associated with myocardial ischemic injury, such as TGF- β signaling and hypoxia on the secretome of cultured fibroblasts isolated from unstressed hearts. Interestingly, we found significant changes in the secretome of fibroblasts treated with TGF- β (Supplementary Figure 7A). On the other hand, the response to hypoxia (1% O₂ versus 20% O₂) was only minor (Supplementary Data 1), which might be due to metabolic adaptation of CF to low oxygen conditions (Janbandhu et al. 2022, doi: 10.1016/j.stem.2021.10.009).

A group of the proteins that were enriched in the miCF secretome were also upregulated after TGF- β stimulation (LOX, FN1, INHBA, POSTN, COL1A1; Supplementary Figure 7B), suggesting that TGF- β signaling contributed to the shift from the cCF secretome to the miCF secretome. Which additional stimuli or combinations of conditions, such as inflammatory cytokines, mechanical stress, or other signaling pathways are involved in the modulation of the secretory process during ischemia needs to be elucidated in future studies. In the revised manuscript we have discussed the new findings (lines 365-376).

3) Were the fibroblasts collected from both the viable and infarcted areas of the heart? As their role and what they secrete is likely vastly different, it would be interesting to stain the hearts for some of the key proteins identified to determine potential spatial differences.

The rapid isolation of cardiac fibroblasts (9 min) required a collagen perfusion method reported by us (Owenier et al. 2020, doi:10.1093/cvr/cvz193) which does not permit to differentiate any longer between infarcted and viable remote areas of the heart. Thus, the miCF fraction we collected from post-MI hearts comprised fibroblasts from all areas.

To address your valid question of potential spatial differences, we made use of a recently published comprehensive spatial multi-omic data set of human myocardial infarction that includes transcriptomes of cardiac fibroblasts from the remote zone and the ischemic zone at different time points post-infarction (Kuppe et al. 2022, doi: 10.1038/s41586-022-05060-x). The use of a transcriptomics data set enabled us to simultaneously analyze a variety of the proteins that we identified to be significantly changed between the miCF secretome and the cCF secretome, without being limited by the availability or specificity of antibodies.

Analysis of the spatial expression data of selected miCF proteins derived from the published human post-MI-heart data set revealed that a group of the proteins are prevalently expressed in fibroblasts located within the ischemic zone (e.g. POSTN, LOX, COL1A1; Supplementary Figure 9A), as one might have expected. Interestingly, some proteins (e.g. FMOD, GDF6, SLIT2; Supplementary Figure 9B) were found to be higher expressed in fibroblasts residing in the remote zone as compared to fibroblasts in the ischemic zone. This suggests that in response to cardiac ischemia, the fibroblasts also in the adjacent non-ischemic remote areas react, which might have been triggered by alterations in the mechanical strain and/or soluble factors (local or circulating). These new results are now reported and discussed in the revised manuscript (lines 299-315).

4) I know that the human data was a previously reported dataset however additional details regarding the type of tissue collected (region of the heart, rejected transplant versus device placement collection, etc.) would be helpful seeing as part of the study is comparing the mouse and human findings. If they are from different, this would be a limitation that should be discussed.

The following experimental details for the human data sets have been included in the revised manuscript (lines 785-790):

“Tissue samples for these data sets were acquired from the left ventricular apex (fresh for CITE-seq and frozen for snRNAseq) and included non-failing controls (rejected for heart transplant but preserved ejection fractions), acute myocardial infarction (within 3 months of MI), and chronic heart failure (ischemic and non-ischemic cardiomyopathy). Tissue from the left ventricular apex was used in all samples to provide consistency in comparing signatures from sample to sample.”

The mouse samples, however, are not restricted to the left ventricular apex which needs to be considered when directly comparing mouse and human data. This limitation is now addressed in the revised manuscript (lines 157-158).